# Spatiotemporal characteristics and optimization strategies of land use and land resource carrying capacity in the three gorges reservoir region (1986–2020)

Hui Li[1,2,3]*, Zhongshan Cui[2], Fuhai Wang[1,3,4], Yunmin Wang[5], Xiaoyuan Zhang[1,3], Honglei Guo[6]

**1** College of Public Management, Chongqing Finance and Economics College, Chongqing, China, **2** School of Management, Xi'an Jiaotong University, Xi'an, China, **3** Chongqing Key Laboratory of Ecological Environment Data Mining and Integrated Application, Chongqing, China, **4** School of Architecture and Urban Planning, Chongqing University, Chongqing, China, **5** School of Economics, Guizhou University, Guizhou, China, **6** School of Geography, Nanjing Normal University,Nanjing, China

* lihui@stu.xjtu.edu.cn

## Abstract

Studying land use changes caused by human economic activities is beneficial for sustainable growth, making it a global research hotspot. In this study, we used Landsat Thematic Mapper images and statistical yearbooks from 1986, 1995, 2000, 2007, 2010, and 2020 to obtain grid data on the land use status of the Three Gorges Reservoir Region (TGRR), from which vector data reflecting socioeconomic information were derived. We introduced models on land use quantitative changes, dynamic indicators, and degree index to investigate spatiotemporal variations in land use in the TGRR over the past 30 years. Classified maps were generated using ARCGIS 10.8, and Landsat TM images were processed for accuracy using supervised classification techniques. Based on the region's status quo and the analytic hierarchy process, we constructed a land resource carrying ability evaluation indicator model considering social, economic, population, and ecological carrying abilities, introducing a mean-square mistake decision-making approach to determine indicator weights. Our results indicate significant changes in land types within the TGRR from 1986 to 2020, with decreases in arable land, forest land, and grassland, while water bodies, building land, and unused land increased. The change rates varied significantly among different land types, reflecting rapid development, especially between 1995 and 2000. Additionally, our analysis delves into the underlying mechanisms driving these changes, providing insights into how different factors influence spatial-temporal evolution of land use and land carrying capacity, crucial for developing optimization strategies aimed at promoting sustainable growth and efficient use of land resources in the TGRR. This study offers a comprehensive analysis of the TGRR's land resource carrying ability, serving as a reference for sustainable land use.

**Data availability statement:** The data underlying the results presented in the study are available from (The basic In this study, data of Landsat Thematic Mapper false-color remote sensing images and statistical yearbooks (social, economic, population, and ecological) for six years (1986, 1995, 2000, 2007, 2010, and 2020) were used.).

**Funding:** This research work was partly supported by the National Social Science Funds of China under Grants No.22XJY006,and the Social Science Research Project of Chongqing Municipal Education Commission under Grants No.24SKGH361.

**Competing interests:** The authors declare no competing interests.

# 1 Introduction

Land is an important resource necessary for human survival [1–3]. Land use/cover change (LUCC) is the main cause of global climate change and is closely related to human activities. Therefore, studying LUCC has gained global emphasis [4–6]. The carrying ability of land resources is an important index for land resource assessment. Prior research has primarily focused on assessing land carrying capacity, which quantifies the population sustainable by regional food production under multidimensional natural, socioeconomic, and institutional constraints. [7,8]. With population and financial growth, accelerating urbanization, and intensification of ecological and environmental issues, the demand for land for regional development is constantly expanding. Thus, research on the carrying capacity of land using population and grain as single indicators, is gradually moving towards a comprehensive indicator system [9–12]. The traditional approach for assessing the carrying ability of land is no longer suitable for regional sustainable growth. The existing land carrying ability is the limit of the scale and intensity of different activities that land resources can carry under certain social, economic, and ecological conditions in a certain period and spatial area [13]. Studying the integrated carrying ability of land resources involves a comprehensive dynamic balance relationship between resources, environment, population, society, economy, and other aspects, thus, reflecting the material, energy, and information flow connections and coordinated development relationships between the natural environment and socioeconomic systems at different regional scales [14–16]. Land carrying capacity research has undergone a paradigm shift from material supply orientation to system coupling analysis. In the theoretical foundation stage, William (1948) defined land carrying capacity as the basic capacity of a region to support living organisms, Terzaghi (1984) constructed a quantitative assessment framework through a mechanistic model, and Fung's model (1990) created a quantitative research paradigm for the man-food nexus.After the 21st century, Liu Junyan (2010) utilized RS-GIS to reveal the spatial and temporal variations of ecological carrying capacity, and Costanza (1997) expanded the classification of ecosystem services to 17 categories, promoting the diversification of the assessment dimensions. Costanza (1997) expanded the classification of ecosystem services to 17 categories and promoted the diversification of assessment dimensions. In recent years, international research has shown three major frontier advances: first, the deep integration of artificial intelligence and big data, such as the EU LandSense platform (2022), which integrates multi-source data to build a global dynamic monitoring model; second, innovation in system dynamics modeling, the MIT team (2023) proposed the "socioecological-technological" coupled model (SET-CCM), which simulates the evolution of carrying capacity by means of the digital twin technology; and third, the development of the ecological carrying capacity by using RS-GIS technology to reveal the spatial and spatial variability of ecological carrying capacity. Third, interdisciplinary theoretical breakthroughs, the Harvard team (2021) proposed a "planetary boundary carrying capacity" framework, which introduces Earth system science into traditional assessment. At the methodological level, we have broken through the static threshold measurement and developed a dynamic feedback mechanism of

"pressure-state-response", and Stanford University (2024) has constructed a digital twin system for global land carrying capacity, which can simulate the impacts of 200 policy scenarios on resource utilization efficiency. These innovations have significantly expanded the global perspective and prediction accuracy of carrying capacity research, providing scientific quantitative support for sustainable development.

In recent years, land use change and its impact on ecosystems and socio-economics have become a global research hotspot. Particularly in the Three Gorges Reservoir Region (TGRR), land use changes have been particularly pronounced due to large-scale infrastructure development and socioeconomic progress. While existing studies have examined land use change trends using multi-source data and long-term time series analysis, assessed land resource carrying capacity through integrated evaluation models, and proposed optimization strategies for sustainable development, these studies often focus on short timeframes or specific types of land changes, lacking systematic integration of multiple influencing factors and concrete, actionable strategies. The innovation of this study lies in: we utilized Landsat imagery and statistical yearbook data from 1986 to 2020 to systematically analyze the spatiotemporal evolution characteristics of land use changes in the Three Gorges Reservoir Region and generated high-precision land use maps; Based on the Analytic Hierarchy Process (AHP) and the mean square deviation decision-making method, we constructed a comprehensive land resource carrying capacity evaluation model that considers social, economic, demographic, and ecological factors; and through an in-depth analysis of the mechanisms by which different driving factors influence land use changes, we proposed targeted optimization strategies aimed at promoting the efficient use of land resources and ecological protection in the Three Gorges Reservoir Area, thereby advancing regional sustainable development..

## 2 Materials and methods

### 2.1 Overview of the study area

The TGRR is a special geographical concept closely related to the Three Gorges Project, and specifically refers to the areas submerged by the construction, storage, and operation of the Yangtze River Three Gorges Project. The TGRR, covering an area of 57,335 km², is geographically located between 106°20′–110°30 E and 29°–31°50′ E in an area combining the Sichuan Basin and the Middle and Lower Yangtze Valley Plains in the middle and lower reaches (Fig 1). The Three Gorges Reservoir Region (TGRR), spanning 22 districts and counties including Chongqing's core urban area, is characterized by a mountainous-dominated geomorphology (76.1% mountainous terrain, 15.3% flatlands) with distinct elevation gradients and a subtropical monsoon climate (annual precipitation: 1000–1800 mm). Its dual identity as a "super mountainous" and "super reservoir" zone exacerbates ecological vulnerabilities, particularly severe soil erosion and landslide risks linked to steep slopes and hydrological fluctuations. Since 1986, accelerated socioeconomic growth driven by Chongqing's municipal status elevation and the Three Gorges Project's impoundment has triggered dramatic land use transitions, intensifying the paradox between abundant total land resources and acute per capita scarcity (notably arable land <0.05 ha/person). These spatiotemporal dynamics, shaped by reservoir inundation, terrain constraints, and anthropogenic pressures, position the TGRR as a critical nexus for analyzing land resource carrying capacity evolution and formulating adaptive optimization strategies under coupled ecological and developmental stresses.

Owing to the unique natural conditions and the establishment of the Three Gorges Project, the ecosystem of the TGRR is unique and fragile that is greatly influenced by its land use status. In the past 30 years, with rapid socioeconomic growth and changes in the natural conditions in the TGRR, the number and spatial features of land use have also changed, leading to significant variations in the land resource carrying capacity [17]. Recently, as ecological migration, urban development, infrastructure construction, and industrial park construction are rapidly increasing in various districts and counties in the TGRR area, land development and construction has consequently increased [18–20]. After the water storage capacity of the reservoir area is complete, the carrying ability of land resources changes significantly, which should be systematically studied urgently. Research on the carrying ability of land resources in the TGRR

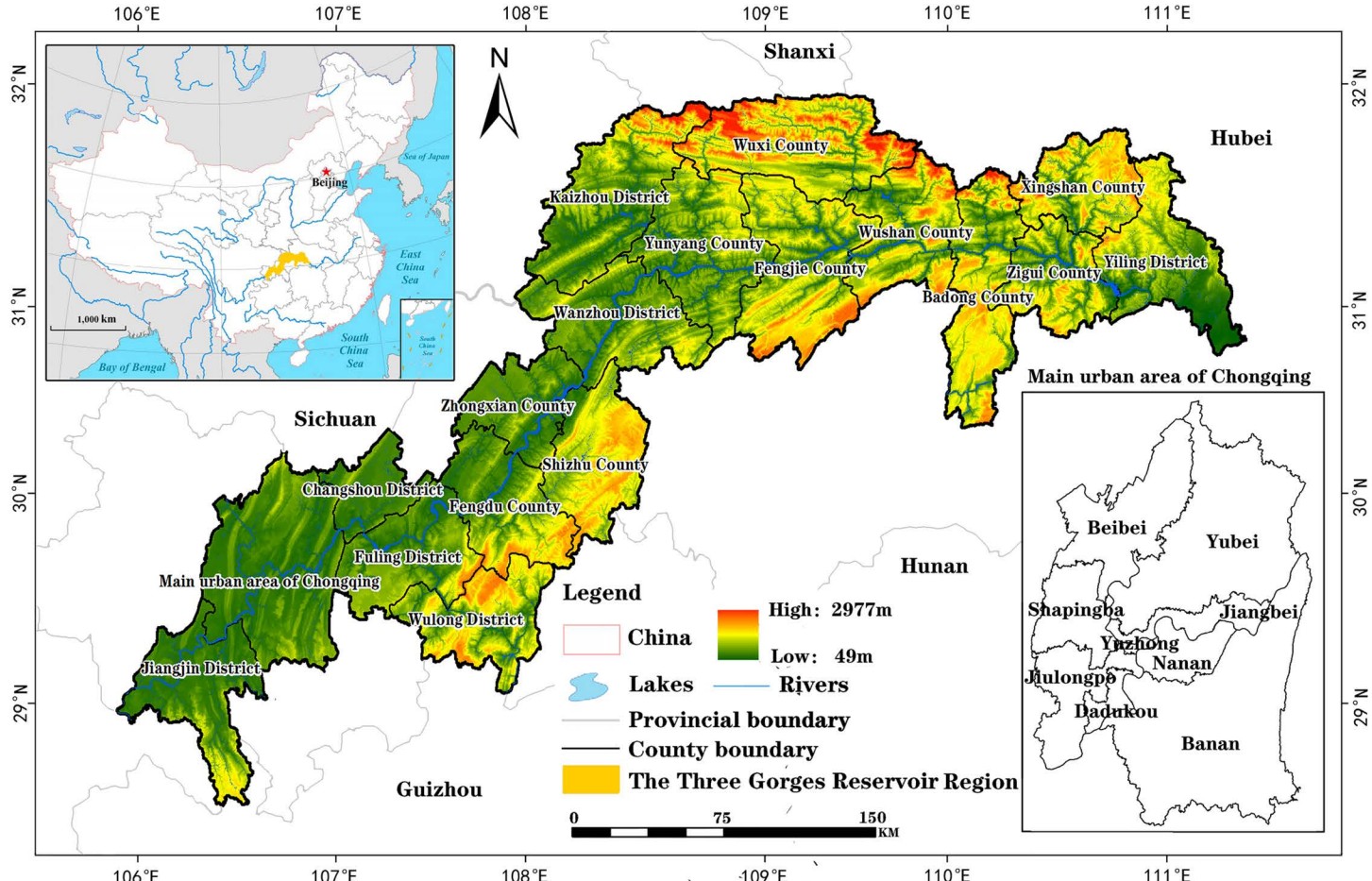

**Fig 1. Location Map of the TGRR.** Note: DEM data is sourced from the Geospatial Data Cloud website (http://www.gscloud.cn/home). Administrative boundary data comes from Resource and Environment Science and Data Center (https://www.resdc.cn/),and the map boundary has not been changed. Cartographic software: ARCGIS 10.8.

can help to scientifically understand the status of population, resources, environment, and economic growth [21–23]; alleviate the contradiction between population growth, economic development, ecology, and resources; and drive the sustainable growth of the TGRR.

## 2.2 Data sources

In this study, Landsat Thematic Mapper (LTM) false-color remote sensing image data (https://search.earthdata.nasa.gov) with an accuracy of 30m*30m were used. administrative division data were obtained from the National Geographic Information Center of China (http://www.ngcc. cn/ngcc/). In addition, population, economic, social, and ecological data were converted from Chongqing Statistical Yearbook (https://tjj.cq.gov.cn/zwgk_233/tjnj/index.html) and Hubei Provincial Statistical Yearbook (https://tjj. hubei.gov.cn/tjsj/sjkscx/tjnj/qstjnj/index.shtml) (1986–2020) and related data such as land use. Land use types were categorized by supervised classification method, and the accuracy comparison results based on 688 randomly selected ground truth sample points showed that the accuracy was 94.52%. Comparison of the accuracy based on 688 randomly selected ground-truth sample points shows that the accuracy is 94.52% and the Kappa coefficients are also highly consistent, thus confirming the accuracy and scientific value of the data.

## 2.3 Research methods

By introducing the dynamic land use index and comprehensive land use index, this study can visually characterize the dynamic features and adapt to multi-temporal comparisons and systematic evaluation of comprehensive benefits. Compared with the entropy weight method (which relies on data discretization) and TOPSIS (which needs to preset ideal solutions), the model in this study is more suitable for the dual objectives of "dynamic monitoring + comprehensive evaluation". Meanwhile, to address the problems that TOPSIS is prone to introduce bias due to the selection of base year in long time series and entropy weighting method is sensitive to extreme values, this study combines AHP and mean square error method to realize dynamic weight adjustment, which effectively improves the comparability and stability of time series. The selected model has been widely used in the field of land science, and its empirical validity at the regional scale provides theoretical and practical support for this study.

**2.3.1 Land use dynamic index model.** To show the dynamic variations in regional land use more intuitively, this study introduced two models, a single land use dynamic level and an integrated land use dynamic degree, to scientifically and objectively investigate the quantity and speed changes of a land use type, and the quantity and speed changes of all land use types in the entire area. Equation 1 shows the calculation formula for a single land use dynamic degree.

$$K = \frac{u_b - u_a}{u_a} \times \frac{1}{T} \times 100\%$$

(1)

where $K$ is the dynamic level of a certain land use type, $u_a$, and $u_b$ represent the number of land use types during the start and end study periods, respectively, and T represents the study period. The $K$ value indicates the rate of variation of a certain land use type in the TGRR [2]. The dynamic level of comprehensive land use can be calculated as follows:

$$Lc = (\sum_{i=1}^{n} \Delta Lu_{i-j}/2 \sum_{i=1}^{n} Lu_i) \times T^{-1} \times 100\%$$

(2)

where $Lu_i$ represents the area of the $i^{th}$ type of land use during the initial research period, $\Delta Lu_{i-j}$ is the area (absolute value) where the $i^{th}$ type of land use was changed to a non-$i^{th}$ type during the research period, namely, into other land use types, $T$ represents the study period, and $Lc$ represents the overall rate of land use change in the area.

**2.3.2 Land use composite index model.** The breadth and depth of land use are mainly reflected in the degree of land use, the natural attributes of land, and the degree of influence of human activities on land [24–26]. Based on the comprehensive analysis method of land use level proposed by Liu et al. and combining with the current situation of TGRR, we divide the land use level into four levels to obtain the grading index of land use degree (Table 1). In this context, "land use degree" refers to the intensity and scope of land development and use, rather than a simple category or type, which reflects the high or low degree of land use. Subsequently, we calculated the land use level index through equation (3) to quantitatively assess the land use in the study area. Such an approach not only refined the land use classification, but also enabled us to understand the impact of different land use patterns more precisely..

$$L = \sum_{i=1}^{n} (A_i \times C_i)$$

(3)

**Table 1. Classification of land use index.**

| Type classification | Unused land level | Land grade for forest land, grass land, and water use level | Agricultural land level | Construction land level |
|---|---|---|---|---|
| Land use type | Unused Land | Forests, Grasslands, and Water | Cultivated Land | Construction Land |
| Graded index | 1 | 2 | 3 | 4 |

where $L$ is the land use indicator of the entire area, $A_i$ represents the land use degree grading indicator of the $i$-level land use type in the region, $C_i$ represents the percentage of the $i$-level land in the entire area of the region, and $N$ is the graded quantity of land use degree. Further, $C_i$ can be calculated as:

$$C_i = CC_i / HJ \tag{4}$$

where $CC_i$ represents the region of the $i^{th}$ level land use type in the region and $HJ$ represents the total land area of the region.

$$\Delta I_{b-a} = I_b - I_a = \left\{ \left( \sum_{i=1}^{n} A_i \times C_{ib} \right) - \left( \sum_{i=1}^{n} A_i \times C_{ia} \right) \right\} \tag{5}$$

where $I_a$ and $I_b$ regional land use indices represent the initial and termination periods, respectively, $A_i$ represents the classification index of the extent of $i$-level land use, and $C_{ib}$ and $C_{ia}$ represent the extent of $i$-level land use during the initial and terminal study periods, respectively. This formula can effectively calculate the degree of land use change in the study area. The selection of indicators (Table 2) for the land resource carrying capacity was based on an analytic hierarchy process (AHP), with weights determined using a mean-square decision-making approach. Each criterion was verified by experts in the field for robustness.

**2.3.3 Assessment index system for land resource carrying capacity.** Considering the actual condition of the TGRR, an evaluation index system was constructed in this study using the AHP approach, which mainly comprises the evaluation target, criterion, and indicator layers. The overall goal was to scientifically and objectively evaluate the land resource carrying capacity of the research area [27,28]. Therefore, we adopted the population, economic, and social carrying ability of land resources, and environmental resource sustainability as reference layers; moreover, 16 indices were selected based on the availability and quantifiability of information to build an assessment system (Table 2).

In the multi-indicator comprehensive evaluation method, weight coefficients of attribute indicators can be determined by subjective weighting approaches (e.g., Gulin, Delphi, and AHP) or objective weighting approaches (e.g., principal component analysis, element analysis, and mean square error method) [29,30]. While subjective methods are widely applied, their results may lack objectivity due to human bias. In contrast, objective methods like the mean square error approach offer higher clarity, computational simplicity, and accuracy in weight determination [31].

Considering the practical context of the Three Gorges Reservoir Region (TGRR), we adopted the mean square deviation approach. This method treats each assessment indicator as a random variable, with dimensionless values of schemes under each indicator as variable realizations. First, the mean square deviation of each indicator is calculated and normalized to derive its weight. The standardized indicator values are then multiplied by their respective weights to obtain subsystem evaluation values (e.g., bearing subsystems), and summed to generate the comprehensive land-bearing capacity score.

First, the mean $E(Aj)$ of the random variable is calculated as:

$$E(Aj) = \frac{1}{n} \sum_{i=1}^{n} yij \tag{6}$$

Second, the mean square deviation of the index set $Aj$ is calculated using the following formula:

$$\sigma(Aj) = \sqrt{\frac{1}{n} \sum_{i=1}^{n} (Y_{ij} - E(A_j))^2} \tag{7}$$

**Table 2. Evaluation system for land resource carrying capacity.**

| Target Layer | Criterion Layer | Indicator Layer | Unit | Nature of Indicator |
|---|---|---|---|---|
| Land carrying capacity | B1. Population carrying capacity of land resources | $X_1$: per-capita construction land | hm² | + |
| | | $X_2$: per-capita cultivated land area | hm²/person | + |
| | | $X_3$: per-capita residential area | hm²/person | + |
| | | $X_4$: population density | person/hm² | + |
| | B2. Economic carrying capacity of land resources | $X_5$: per-capita GDP | CNY/person | + |
| | | $X_6$: per-capita total retail sales of social consumer goods | CNY/person | + |
| | | $X_7$: Contribution rate of the tertiary industry to GDP | % | + |
| | | $X_8$: Average fixed assets investment | CNY/hm² | + |
| | B3. Social carrying ability of land resources | $X_9$: per-capita public service facility land area | hm²/person | + |
| | | $X_{10}$: natural population growth rate | % | − |
| | | $X_{11}$: population employment rate | % | + |
| | | $X_{12}$: urbanization rate | % | + |
| | B4. Ecological carrying ability of land resources | $X_{13}$: forest coverage rate | % | + |
| | | $X_{14}$: per-capita public green space area | hm²/person | + |
| | | $X_{15}$: sewage treatment rate | % | + |
| | | $X_{16}$: Comprehensive utilization rate of industrial solid waste | % | + |

Note: "+" means a positive index, and "-" a negative index.

Third, the weight value of indicator $Aj$ was calculated as follows:

$$W(Aj) = \frac{\sigma(Aj)}{\sum\limits_{j=1}^{m} \sigma(Aj)}$$

(8)

Finally, the comprehensive evaluation value was calculated as:

$$D_i(W) = \sum_{j=1}^{m} yijw(Aj)$$

(9)

The study adopts a systematic framework for assessing land carrying capacity, which is divided into the following three main steps: in the first step, a three-level indicator system containing four first-level indicators (demographic, economic, social, and ecological carrying capacity) is constructed by acquiring and pre-processing data in the study area (including determining the scope, unifying the resolution of the data, and extracting the land-use information); in the second step, the current land-use situation is analyzed, the Calculate the standard values of land use changes (e.g., K, Lc, L) using specific formulas, and classify and quantify the land use data in order to clarify the changes in land use; in the third step, at the stage of comprehensive land carrying capacity assessment, standardize the indicators of the same guideline layer through the maximum-minimum value method, then apply the hierarchical analysis method (AHP) to determine the weights of each evaluation indicator, and finally, based on the The standardized indicator values and weights are then used to determine the weights of each evaluation indicator, and finally the integrated evaluation value (Di(W)) is calculated based on the standardized indicator values and weights to arrive at the final conclusion about the land carrying capacity. This framework provides a comprehensive assessment of the land carrying capacity of the study area through a multi-step systematic analysis. The framework diagram of the research methodology is presented in Fig 2.

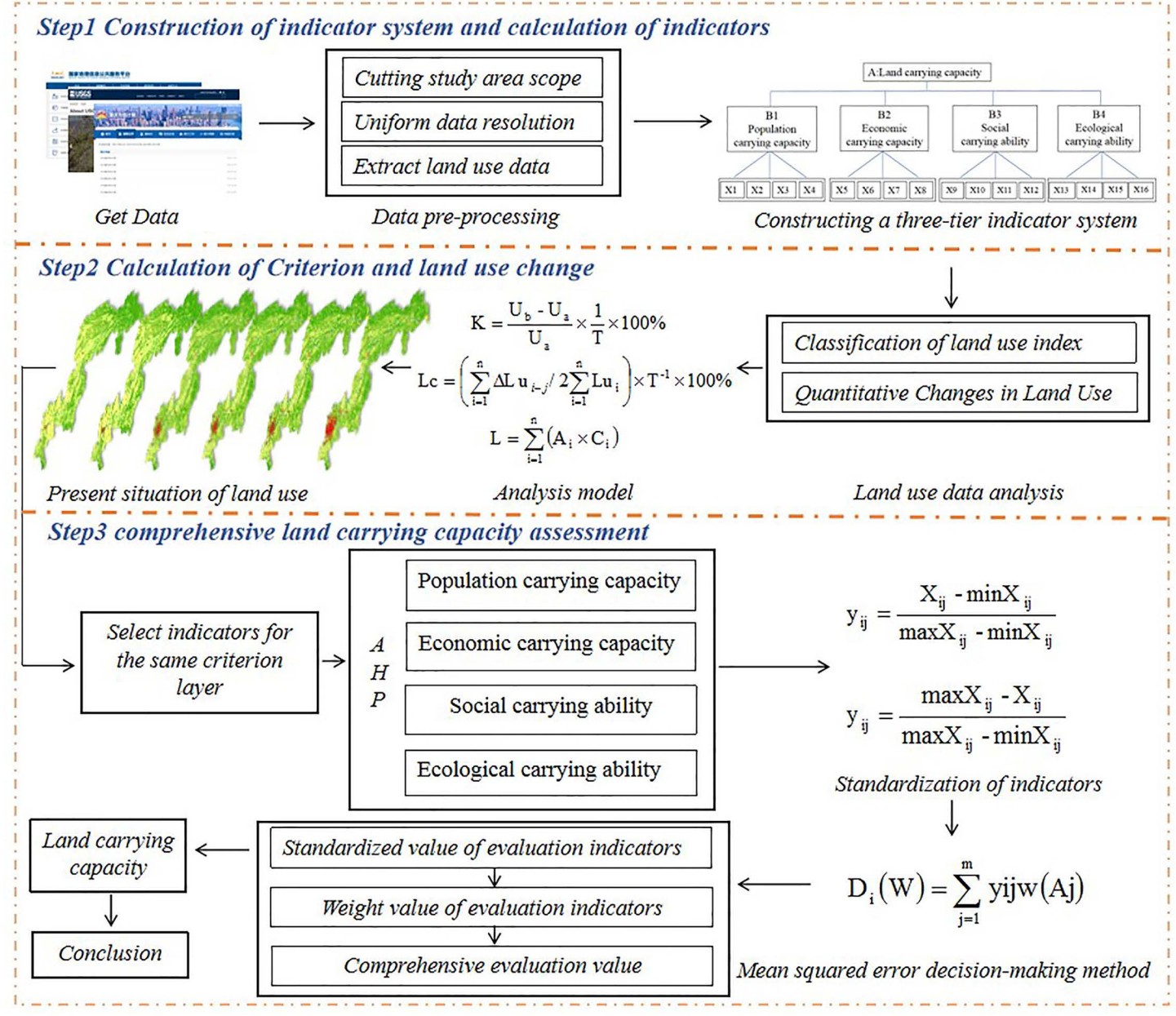

**Fig 2. Research framework diagram.** Note: The basemap was obtained from the Geospatial Data Cloud (http://www.gscloud.cn/home), and the map boundary has not been changed.Cartographic software:ArcGIS 10.8.

## 3 Results

### 3.1 Quantitative changes of land use types

Spatial information technology was used to statistically analyze land use status in the TGRR from 1986 to 2020. The results are presented in Table 3. Using the mapping function of ARCGIS 10.8 software, a classification map of the land use status was created, as shown in Fig 3. Figure 3 shows the classified maps of land use change over the years, clearly

**Table 3. Statistical table of land use types and areas in the TGRR from 1986 to 2020 (km², %).**

| Land Use Types | 1986 | 1995 | 2000 | 2007 | 2010 | 2020 |
|---|---|---|---|---|---|---|
| Cultivated Land | 22141.89 | 21920.95 | 21840.97 | 21699.71 | 21564.25 | 21180.44 |
| | 38.62 | 38.23 | 38.09 | 37.85 | 37.61 | 37.22 |
| Forest | 31879.08 | 31720.26 | 31471.42 | 31408.35 | 31380.23 | 31136.10 |
| | 55.60 | 55.32 | 54.89 | 54.78 | 54.73 | 54.43 |
| Grassland | 2306.04 | 2286.12 | 1408.92 | 1405.73 | 1398.50 | 1356.62 |
| | 4.02 | 3.99 | 2.46 | 2.45 | 2.44 | 2.43 |
| Water | 764.87 | 964.14 | 1323.05 | 1432.63 | 1484.62 | 1573.78 |
| | 1.33 | 1.68 | 2.31 | 2.50 | 2.59 | 2.64 |
| Construction Land | 234.59 | 434.71 | 1273.61 | 1372.28 | 1490.63 | 2071.87 |
| | 0.41 | 0.76 | 2.22 | 2.39 | 2.60 | 3.25 |
| Unused Land | 8.80 | 9.09 | 17.30 | 16.57 | 17.04 | 16.46 |
| | 0.02 | 0.02 | 0.03 | 0.03 | 0.03 | 0.03 |
| Sum | 57335.27 | 57335.27 | 57335.27 | 57335.27 | 57335.27 | 57335.27 |
| | 100 | 100 | 100 | 100 | 100 | 100 |

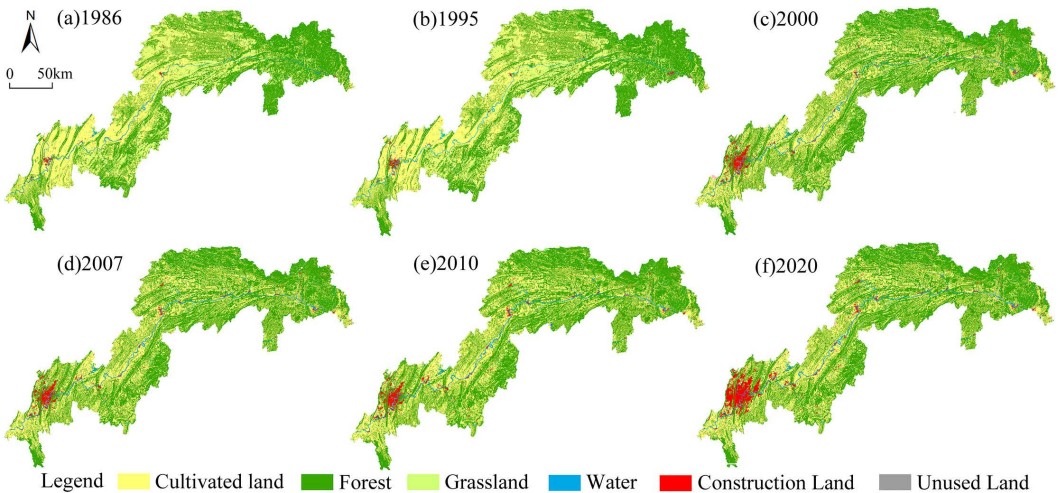

Legend: Cultivated land · Forest · Grassland · Water · Construction Land · Unused Land

**Fig 3. Land use status map of the TGRR from 1986 to 2020.** Note: The basemap was obtained from the Geospatial Data Cloud (http://www.gscloud.cn/home), and the map boundary has not been changed. Cartographic software: ArcGIS 10.8.

indicating the spatial transformation of the landscape in the TGRR. Table 3 presents the area and proportion of different land use types over the years, providing a clear view of the changes in the quantities of various land uses within the study area.

The data of the previous 30 years presented in Table 3 and Fig 3 indicate that the overall land use structure of the TGRR has been stable, with forest land and arable land being the dominant land use types. The distribution of arable land in the middle and upper reaches of the TGRR was relatively concentrated; moreover, the forest land was concentrated in the first section of the TGRR and the ranges of Mingyue, Zhongliang, Jinyun and Tongluo. Grasslands, as the third terrestrial land use type showed a sporadic distribution pattern in various areas of the TGRR, while construction land showed a relatively concentrated distribution, such as in the core urban region of Chongqing. In terms of quantitative changes, the

land use types that showed an increasing trend during the research period were building land, water areas, and unused land, whereas cultivated land, forest land, and grassland showed a decreasing trend. Building land had the largest land category in the TGRR, increasing from 0.41% to 2.84%, with a net increase of 1837.28 km². This increase was because of the rapid growth of the social economy in the TGRR, which has a large amount of construction land. Cultivated land was the largest land type in the TGRR, which decreased from 38.62% to 37.22%, with a net decrease of 961.45 km².

Based on the data analyzed in Table 3, the construction land in the Three Gorges Reservoir Region (TGRR) showed a significant expansion trend from 1995 to 2000, with the area increasing from 434.71 km² to 1,273.61 km², an increase of 193%. This phenomenon is closely related to the national major strategic projects and policy regulation during the same period. Specifically, the Three Gorges Water Conservancy Hub Project was officially launched in 1994, which directly triggered the large-scale resettlement of immigrants in the reservoir area and the acceleration of the urbanization process, prompting a surge in the demand for land for infrastructure, housing and public services. At the same time, the promulgation and implementation of the Regulations on Returning Cultivated Land to Forests in 1998 may guide the conversion of cultivated land and grassland to forest land through the mechanism of ecological compensation, resulting in a 45% reduction in the area of grassland in the same period (2,286.12 km² to 1,408.92 km²), whereas the forest coverage rate remained relatively stable (55.32% to 54.89%), which embodies the dual regulation of the land-use structure by policy. This reflects the dual regulation of the land use structure by the policy. It is worth noting that the water area increased continuously from 964.14 km² in 1995–1,573.78 km² in 2020, and the stage-by-stage growth nodes (2003, 2006, and 2008) coincided with the water storage cycle of the Three Gorges Project, which confirms the transformation of the natural geographic pattern by the major projects. 167.78 km² to 34.85 km²) after 2000, which may be related to the binding effect of ecological protection policies such as the National Ecological Functional Zoning (2008).

## 3.2 Land use dynamics and accuracy assessment in the TGRR

### 3.2.1 Speed of land use variation in the TGRR.
Using Equations 1 and 2, the rate of land use variation in the TGRR was analyzed, and the single and comprehensive dynamic levels of every region over the past 30 years were obtained, as shown in Table 4. In this study, the approach for handling transitions between different types of land involved modeling the conversion between land types using temporal change detection algorithms in ARCGIS, and cross-referencing these transitions with historical land use maps for verification.

On analyzing the variation rate of different land use types, we found that building land showed the fastest rate of land use change during the research period. Particularly, from 1995 to 2000, the change rate of construction land was the most significant, with a positive land use change rate of up to 38.60%. Moreover, the land types with the highest relative rates of land use change during 1986–1995, 1995–2000, 2007–2010, and 2010–2020 were construction land, with change rates of 9.47%, 38.59%, 2.87%, and 6.22%, respectively, while the land type with the highest relative rate of land use variation during 2000–2007 was water, with a change rate of 1.18%.

The integrated dynamic degree of land use indicated that the period from 1995 to 2000 exhibited the highest rate of land use variation in the research area, with a comprehensive dynamic degree of 0.42%. Therefore, in the past 30 years,

**Table 4. Dynamic degree of land use of various types in the TGRR from 1986 to 2020 (Unit: %).**

| Period of Time | Cultivated Land | Forest | Grassland | Water | Construction Land | Unused Land | Comprehensive Dynamic Degree |
|---|---|---|---|---|---|---|---|
| 1986-1995 | −0.11 | −0.06 | −0.10 | 2.89 | 9.47 | 0.36 | 0.07 |
| 1995-2000 | −0.07 | −0.16 | −7.67 | 7.45 | 38.59 | 18.05 | 0.42 |
| 2000-2007 | −0.09 | −0.03 | −0.03 | 1.18 | 1.10 | −0.60 | 0.05 |
| 2007-2010 | −0.21 | −0.03 | −0.17 | 1.21 | 2.87 | 0.95 | 0.09 |
| 2010-2020 | −0.26 | −0.14 | −0.04 | 0.49 | 6.22 | −0.11 | 0.17 |

land use in the TGRR was most active and rapid during this period, mainly because of the rapid variation rate of construction land, water bodies, and unused land, which showed an increasing trend. Moreover, land types, such as cultivation, forest, and grassland, also exhibited relatively high rates of change.

**3.2.2 Accuracy assessment of land use classification maps and model performance.** In this study, we place great emphasis on the accuracy assessment of land use classification, as it is a key indicator of the reliability and effectiveness of our classification method. To ensure that readers can fully understand the reliability of land classification over time, we conducted a comprehensive accuracy assessment of the classification results using the confusion matrix validation method. Specifically, we used the confusion matrix to calculate the user accuracy, producer accuracy, overall accuracy, and Kappa Coefficient for five time intervals (1986–2020) in the study area. These indicators can intuitively reflect the consistency between the classification results and actual observed data. By comparing the classification results over different time periods, we found that although there were some fluctuations, the overall accuracy remained at a high level. For instance, in the five classification periods, we achieved an average overall accuracy of 92.14% and an average Kappa Coefficient of 88.47%, indicating that our classification method has high stability and reliability. These evaluation results not only prove that our classification method is accurate and reliable but also provide readers with an important basis for understanding the reliability of land classification changes over time. Detailed data are presented in Table 5.

**3.2.3 Land use degree in the TGRR.** To study the extent of land use in the TGRR, further understanding can be gained on the development degree and driving force system of land use variation [32–34], and whether land use in the TGRR is in a developmental state can be determined. Using vector data of land use status in the TGRR from 1986, 1995, 2000, 2007, 2010, and 2020, based on changes in the number of land use types, the calculation formula for the land use degree was used to obtain the integrated indicators shown in Table 6.

Table 6 shows that over the past 30 years, the land type with the highest extent of land use in the TGRR was arable land, while the land type with the lowest degree of land use was unused land. The land use level of construction land and water areas showed an increasing tendency every year, and the land types with a decreasing trend in the integrated indicator of land use level were mainly forest land, cultivated land, and grassland; overall, the variation in land use level of unused land was not significant.

The variations in the land use level index over the past 30 years show that from 1986 to 1995, variations in the land use level index were the largest, with a land use degree change of 0.0278, while from 1995 to 2000, variations in the land use level index were the smallest, with a variation degree of 0.0010 in the land use level. Overall, the variations in land use degree in each period were positive, showing that land use in the TGRR has always undergone development, but there are certain differences in the development degree in the time dimension.

## 3.3 Verification of land use classification results

Using Equations 6 and 7, that is, the range transformation method, standardized calculations were conducted on the raw data of 16 evaluation indices of land resource carrying ability. The acquired standardized values are shown in Table 7.

**Table 5. Accuracy Validation Statistics Table for Land Use Classification Confusion Matrix.**

| Year/Accuracy | User Accuracy (Arithmetic Mean) | Producer Accuracy (Arithmetic Mean) | Overall Accuracy | Kappa Coefficient |
|---|---|---|---|---|
| 1986-1995 | 91.12% | 90.28% | 91.64% | 88.55% |
| 1995-2000 | 88.69% | 89.27% | 91.51% | 87.26% |
| 2000-2007 | 90.83% | 91.61% | 93.13% | 89.14% |
| 2007-2010 | 87.95% | 88.62% | 92.59% | 89.47% |
| 2010-2020 | 89.49% | 90.83% | 91.84% | 87.93% |

**Table 6. Integrated indicator of land use level of various types in the TGRR from 1986 to 2020.**

| Year | Cultivated Land | Forest | Grassland | Water | Construction Land | Unused Land | Regional comprehensiveness | △Ib-a |
|------|-----------------|--------|-----------|-------|-------------------|-------------|----------------------------|-------|
| 1986 | 1.1585 | 1.1120 | 0.0804 | 0.0267 | 0.0164 | 0.0002 | 2.3942 | 0.0031 |
| 1995 | 1.1470 | 1.1065 | 0.0797 | 0.0336 | 0.0303 | 0.0002 | 2.3973 | 0.0278 |
| 2000 | 1.1428 | 1.0978 | 0.0491 | 0.0462 | 0.0889 | 0.0003 | 2.4251 | 0.0010 |
| 2007 | 1.1354 | 1.0956 | 0.0490 | 0.0500 | 0.0957 | 0.0003 | 2.4261 | 0.0017 |
| 2010 | 1.1283 | 1.0946 | 0.0488 | 0.0518 | 0.1040 | 0.0003 | 2.4278 | 0.0091 |
| 2020 | 1.1166 | 1.0885 | 0.0487 | 0.0528 | 0.1299 | 0.0003 | 2.4369 | — |

**Table 7. Standardized values of land resource carrying capacity evaluation indicators in the TGRR.**

| index | 1986 | 1995 | 2000 | 2007 | 2010 | 2020 |
|-------|------|------|------|------|------|------|
| $X_1$ | 0.0000 | 0.0659 | 0.7380 | 0.7621 | 0.8047 | 1.0000 |
| $X_2$ | 1.0000 | 0.5360 | 0.2919 | 0.1895 | 0.1057 | 0.0000 |
| $X_3$ | 0.0000 | 0.1876 | 0.4934 | 0.3128 | 0.5805 | 1.0000 |
| $X_4$ | 1.0000 | 0.6441 | 0.3934 | 0.0000 | 0.2764 | 0.2232 |
| $X_5$ | 0.0000 | 0.0064 | 0.0129 | 0.1768 | 0.4748 | 1.0000 |
| $X_6$ | 0.0000 | 0.0391 | 0.1127 | 0.3437 | 0.6252 | 1.0000 |
| $X_7$ | 0.0000 | 0.4510 | 0.8580 | 1.0000 | 0.6194 | 0.8521 |
| $X_8$ | 0.0000 | 0.0143 | 0.0466 | 0.1972 | 0.5730 | 1.0000 |
| $X_9$ | 0.0000 | 0.0853 | 0.2793 | 0.5513 | 0.7778 | 1.0000 |
| $X_{10}$ | 0.5000 | 0.0000 | 0.3030 | 1.0000 | 0.5591 | 0.5952 |
| $X_{11}$ | 0.5454 | 1.0000 | 0.7207 | 0.0000 | 0.0993 | 0.4056 |
| $X_{12}$ | 0.0000 | 0.1698 | 0.3626 | 0.7547 | 0.8806 | 1.0000 |
| $X_{13}$ | 0.0000 | 0.0190 | 0.8558 | 0.8459 | 0.8413 | 1.0000 |
| $X_{14}$ | 0.0000 | 0.0288 | 0.0648 | 0.3122 | 0.7134 | 1.0000 |
| $X_{15}$ | 0.0000 | 0.0784 | 0.2821 | 0.3549 | 0.7772 | 1.0000 |
| $X_{16}$ | 0.0000 | 0.4089 | 0.7714 | 0.8188 | 0.9366 | 1.0000 |

The weight values of the assessment indicator system were calculated according to the aforementioned steps for calculating indicator weights, combined with the dimensionless values of the 16 evaluation indicators in this study to obtain the specific weight values of each index, as displayed in Table 8.

This study determines weights using the mean square deviation method within the objective weighting method (Wang Mingtao, 1999), whose core principle lies in reflecting the information content and variability of the data itself: each indicator is treated as a random variable, with its value being the standardized value of the evaluation unit. The spatial variability of each indicator across all units is quantified by calculating the mean square deviation of its values across all units; the greater the variability (the larger the mean square deviation), indicates that the indicator has a stronger ability

**Table 8. Weight values of evaluation indicators for land resource carrying ability in the TGRR based on the mean square error method.**

| Indicator | Weight | Indicator | Weight | Indicator | Weight | Indicator | Weight |
|-----------|--------|-----------|--------|-----------|--------|-----------|--------|
| $X_1$ | 0.0673 | $X_5$ | 0.0637 | $X_9$ | 0.0635 | $X_{13}$ | 0.0731 |
| $X_2$ | 0.0586 | $X_6$ | 0.0630 | $X_{10}$ | 0.0532 | $X_{14}$ | 0.0665 |
| $X_3$ | 0.0612 | $X_7$ | 0.0585 | $X_{11}$ | 0.0604 | $X_{15}$ | 0.0633 |
| $X_4$ | 0.0565 | $X_8$ | 0.0645 | $X_{12}$ | 0.0654 | $X_{16}$ | 0.0611 |

to distinguish the carrying capacity of different regions and contains a larger amount of information. After normalization, it is assigned a higher weight, ensuring that the weight allocation is derived from the intrinsic objective attributes of the data and avoids subjective interference. This weighting method based on spatial variability directly determines the sensitivity of the evaluation results to key differential indicators: high-weight indicators (i.e., indicators with significant spatial variability) have a decisive influence on the spatial distribution pattern of the final comprehensive evaluation value and carrying capacity grade. In this context, indicators with high spatial variability often correspond to core limiting factors or key advantageous factors affecting the reservoir area's carrying capacity. The application of the standard deviation method to assign higher weights to such indicators enables evaluation results to more accurately identify and focus on these key drivers of regional differences, thereby objectively revealing the primary causes of differentiation. Additionally, this objective method, which calculates weights based on actual data, enhances the reproducibility and stability of evaluation results.

The values of different evaluation indices for the land resource carrying ability of the TGRR were determined using the mean square deviation weighting method (Table 9).

### 3.4 Integrated assessment of land resource carrying capacity in the TGRR

Based on the actual conditions of the TGRR, combined with the relevant calculation formulas and models introduced in this study, through objective calculations and tests, the evaluation results of the indicators corresponding to the social, economic, population, and ecological carrying abilities of land resources in the TGRR were obtained, as shown in Figs 4-7.

According to the results of the analysis diagram of the population carrying system (Fig 4), all evaluation index factors in the population carrying ability of land resources were fluctuating. The per capita building land and per capita residential land use area showed an overall upward trend. From 1986 to 2007 and 2010–2020, the population density in the TGRR displayed a decreasing trend, but from 2007 to 2010, an increasing trend was observed. In the past 30 years, due to the continuous decrease in arable land area and increasing population in the research area, the per capita arable land area showed a decreasing trend, reaching its lowest value in 2020.

Fig 5 shows that, in the economic carrying system of land resources, the three indicators of per capita gross domestic product (GDP), per capita total retail sales of social consumer products, and per capita fixed asset investment increased

**Table 9. Evaluation values of evaluation indicators based on mean square error method.**

| Evaluation value | 1986 | 1995 | 2000 | 2007 | 2010 | 2020 |
|---|---|---|---|---|---|---|
| $X_1$ | 0.0000 | 0.0044 | 0.0497 | 0.0513 | 0.0542 | 0.0673 |
| $X_2$ | 0.0586 | 0.0314 | 0.0171 | 0.0111 | 0.0062 | 0.0000 |
| $X_3$ | 0.0000 | 0.0115 | 0.0302 | 0.0192 | 0.0356 | 0.0612 |
| $X_4$ | 0.0565 | 0.0364 | 0.0222 | 0.0000 | 0.0156 | 0.0126 |
| $X_5$ | 0.0000 | 0.0004 | 0.0008 | 0.0113 | 0.0303 | 0.0637 |
| $X_6$ | 0.0000 | 0.0025 | 0.0071 | 0.0217 | 0.0394 | 0.0630 |
| $X_7$ | 0.0000 | 0.0264 | 0.0502 | 0.0585 | 0.0362 | 0.0499 |
| $X_8$ | 0.0000 | 0.0009 | 0.0030 | 0.0127 | 0.0370 | 0.0645 |
| $X_9$ | 0.0000 | 0.0054 | 0.0177 | 0.0350 | 0.0494 | 0.0635 |
| $X_{10}$ | 0.0266 | 0.0000 | 0.0161 | 0.0532 | 0.0297 | 0.0317 |
| $X_{11}$ | 0.0330 | 0.0604 | 0.0436 | 0.0000 | 0.0060 | 0.0245 |
| $X_{12}$ | 0.0000 | 0.0111 | 0.0237 | 0.0493 | 0.0576 | 0.0654 |
| $X_{13}$ | 0.0000 | 0.0014 | 0.0626 | 0.0618 | 0.0615 | 0.0731 |
| $X_{14}$ | 0.0000 | 0.0019 | 0.0043 | 0.0208 | 0.0475 | 0.0665 |
| $X_{15}$ | 0.0000 | 0.0050 | 0.0179 | 0.0225 | 0.0492 | 0.0633 |
| $X_{16}$ | 0.0000 | 0.0250 | 0.0472 | 0.0501 | 0.0573 | 0.0611 |

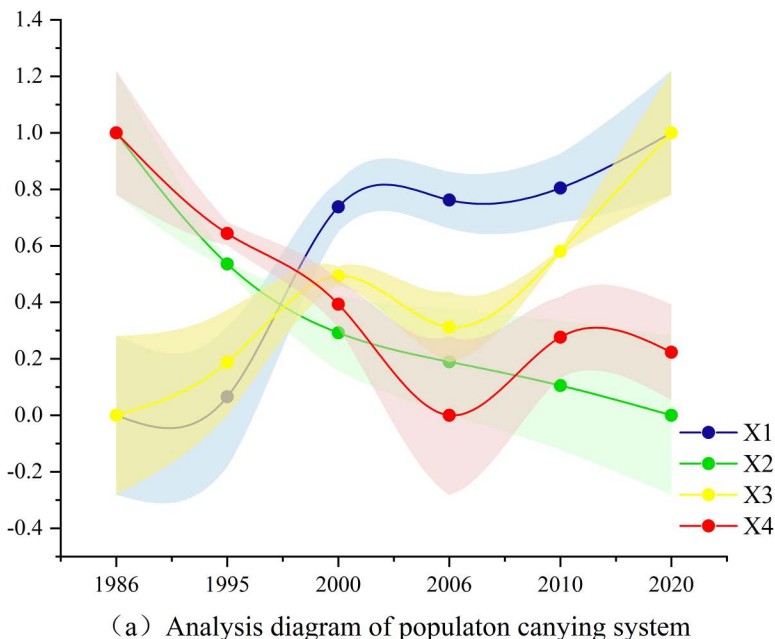

（a）Analysis diagram of populaton canying system

**Fig 4. Analysis diagram of population carrying system.**

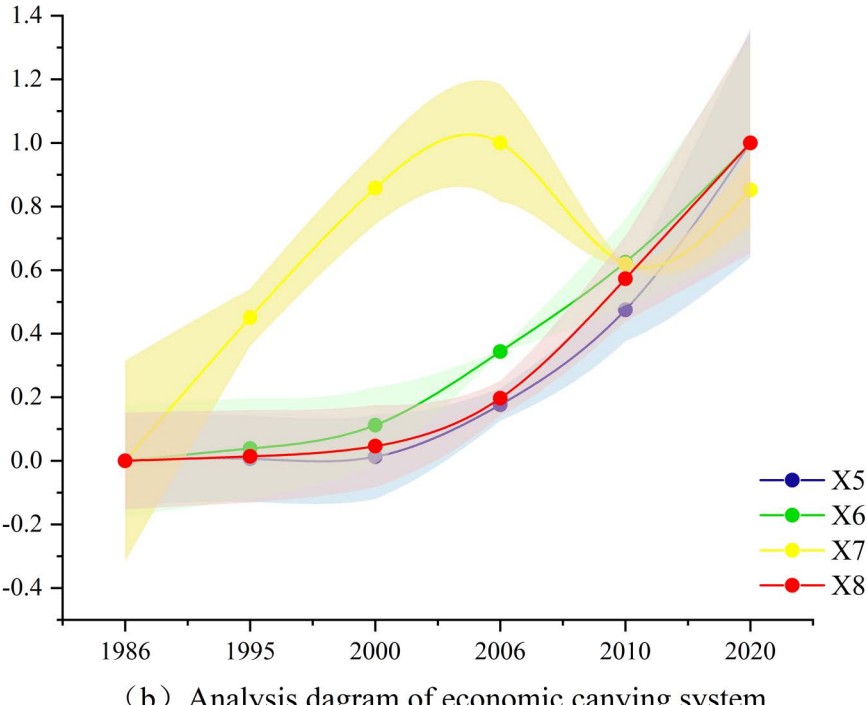

（b）Analysis dagram of economic canying system

**Fig 5. Analysis diagram of economic carrying system.**

rapidly, and the evaluation indicator values were relatively close. The contribution rate of the tertiary industry to GDP showed an upward trend and a significant increase from 1986 to 2007, mainly because of the national reform and opening-up policy and the rapid economic development of Chongqing after being directly under the central government. From 2007 to 2010, there was a downward trend related to the impact of the 2008 financial crisis and the emphasis on industrial development policies in Chongqing at this stage. It also increased from 2010 to 2020.

The results of the analysis diagram of the social carrying system (Fig 6) indicated that in the social carrying ability system of land resources, with the rapid development of Chongqing's direct administration and economy, the per capita area of public service facilities and urbanization rate showed a rapid increasing trend. In terms of population employment rate, there was an upward trend from 1986 to 1995, and a decrease from 1995 to 2007, partly due to the expansion of Chongqing's jurisdiction and the increase in population. An upward trend was also observed from 2007 to 2020. The natural population growth rate decreased from 1986 to 1995, increased from 1995 to 2007, decreased from 2007 to 2010, and slowly increased from 2010 to 2020.

Further, according to the analysis diagram of the ecological carrying system (Fig 7) in the ecological carrying ability system of land resources in the TGRR, the forest coverage rate, per capita public green space area, sewage treatment rate, and integrated use rate of industrial solid waste all showed an upward trend with significant fluctuations. The forest coverage rate increased significantly from 1995 to 2000, which is related to the national policy of returning farmland to forests.

Based on the trend results of evaluation values of land resource carrying ability of each subsystem in the TGRR (Fig 8), in the past 30 years, the social, economic, population, and ecological carrying abilities showed varying rates of increase. The ecological carrying ability had the largest fluctuation compared to the economic carrying capacity, mainly because of the unprecedented rapid economic growth of Chongqing since its direct administration in 1997. With economic growth, more resources have been invested in ecological environmental protection and governance, and

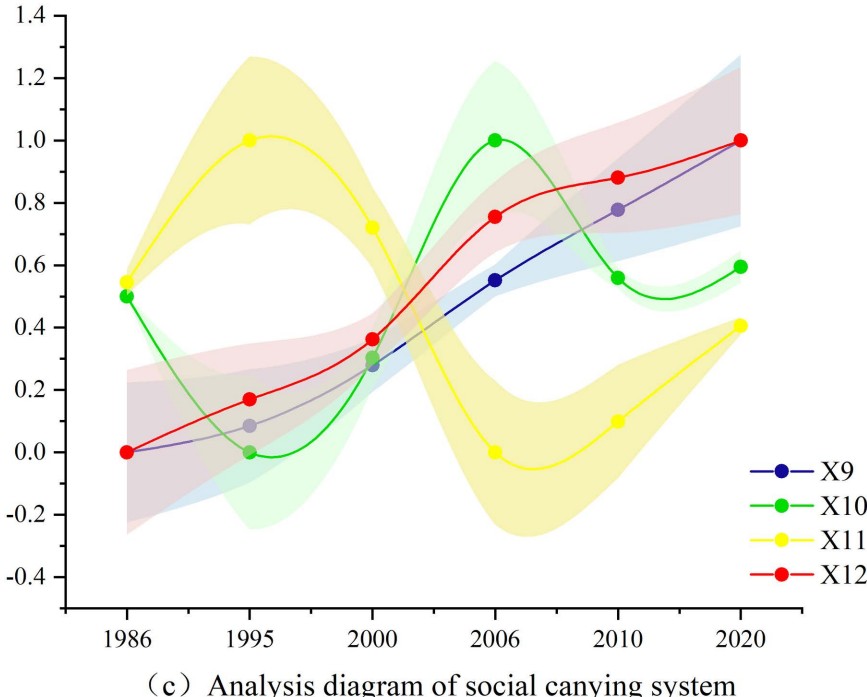

（c）Analysis diagram of social canying system

**Fig 6. Analysis diagram of social carrying system.**

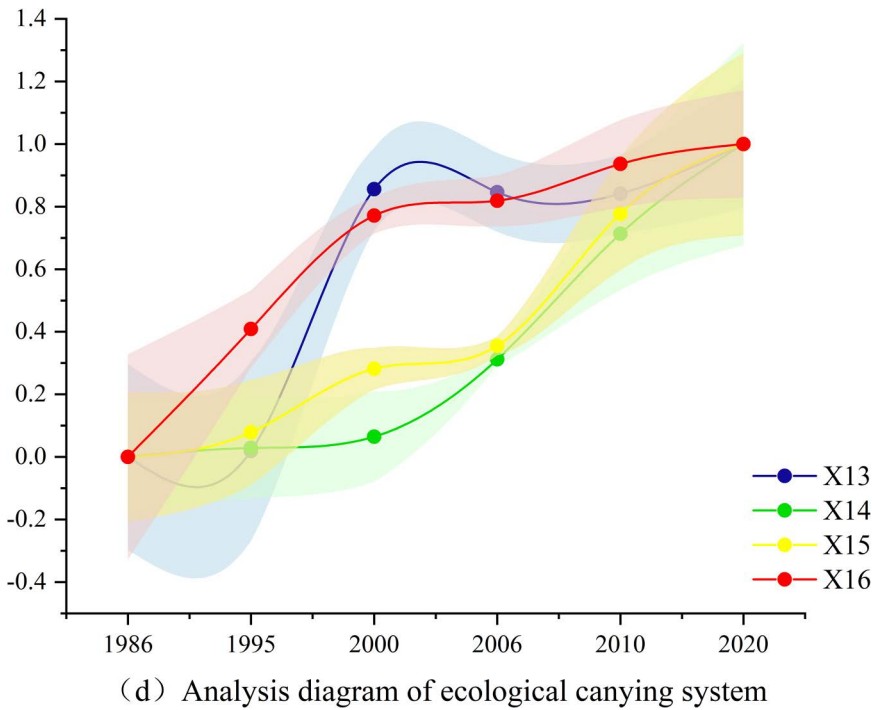

（d）Analysis diagram of ecological canying system

**Fig 7. Analysis diagram of Ecological carrying system.**

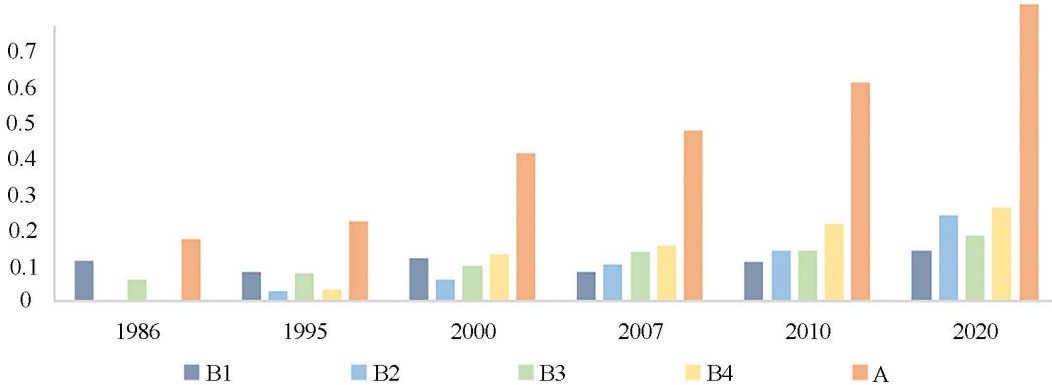

**Fig 8. Trend of evaluation values of land resource carrying ability of each subsystem in the TGRR.** Note: A: Land carrying ability; B1: Population carrying ability of land resources; B2: Economic carrying ability of land resources; B3: Social carrying ability of land resources; B4: Ecological carrying ability of land resources.

progress has been made in the construction of an ecological civilization. The social and population carrying capacities of the research area showed an upward trend; however, the rate was not significant. On analyzing each period, we found that the overall situation of the four individual carrying capacities in the TGRR before 2000 was as follows: population carrying capacity was superior to social carrying capacity, and ecological carrying capacity was superior to economic carrying ability.

For an integrated and intuitive discussion of the carrying ability of land resources, definitive criteria should be established. After expert views and classification standards were proposed by various researchers, this study categorized the sustainable use of land resources into five levels, as displayed in Table 10.

The integrated assessment value of land resource carrying ability in the TGRR was obtained based on the calculation formula and land carrying ability grading evaluation standards, as shown in Fig 9.

Looking at the temporal dimensions (Fig 9):

(1) The evaluation value of the integrated carrying ability of land resources in the study area in 1986 was 0.1746, showing that the degree of land resource carrying ability was low. A discussion of the various subsystems of land resource carrying ability in the TGRR shows that the evaluation values of land resource economic carrying ability and land resource ecological carrying ability in that year were close to zero, while the assessment value of land resource population carrying ability remains at a level of 0.1, reflecting the uneven development between the subsystems of land resource carrying ability in each study area, low level of regional economic growth, and poor ecological environment conditions.

(2) The assessment value of the integrated carrying ability of land resources in the TGRR in 1995 was 0.2241, indicating that the land resource carrying ability was at a "critical" level. The discussion on various subsystems of land resource carrying ability shows that the evaluation values of land resource economic carrying ability and land resource ecological carrying ability in that year remained around 0.03, while the evaluation values of land resource population and land resource social carrying abilities were approximately 0.08. This indicated that after nearly five years of development, the evaluation of land resource economic and ecological carrying abilities in the TGRR showed a potential for improvement. However, there is a gap between the population and social carrying ability of land resources.

(3) The assessment value of the integrated carrying ability of land resources in the TGRR in 2000 was 0.4134, indicating that the land resource carrying ability was at the "initial" level. The evaluation values of the population, social, and ecological carrying abilities of land resources in that year remained approximately 0.1. Notably, the economic carrying ability of land resources increased to 0.0631. That was because the direct administration of Chongqing led to significant economic and social growth in most of the reservoir area, resulting in an increasing trend in all subsystems. On analyzing the various land resource carrying systems of land resources, ecological carrying ability of land resources in the study area showed some improvement, while the speed of the economic development in the TGRR was significant. Overall, the assessment of the integrated carrying ability of land resources in the study area at this stage can be increased significantly. However, there is still significant room for improvement in the balance between various land resource subsystems.

(4) In 2007, the evaluation value of the integrated carrying ability of land resources in the TGRR was 0.4784, indicating that the land resource carrying capacity was still at its "initial" level. The population carrying capacity of the land resources during this period showed a downward trend. This could be possibly because the significant increase in population in Chongqing decreased the values of indicators, such as per capita arable land area and per capita residential area in the TGRR, ultimately decreasing the population carrying ability value of land resources. Additionally,

**Table 10. Criteria to evaluate the degree of carrying capacity land resources.**

| Comprehensive Value | <0.2 | 0.2-0.4 | 0.4-0.6 | 0.6-0.8 | >0.8 |
|---|---|---|---|---|---|
| Criteria | Low carrying capacity | Critically carrying capacity | Initially carrying capacity | Basically carrying capacity | High carrying capacity |

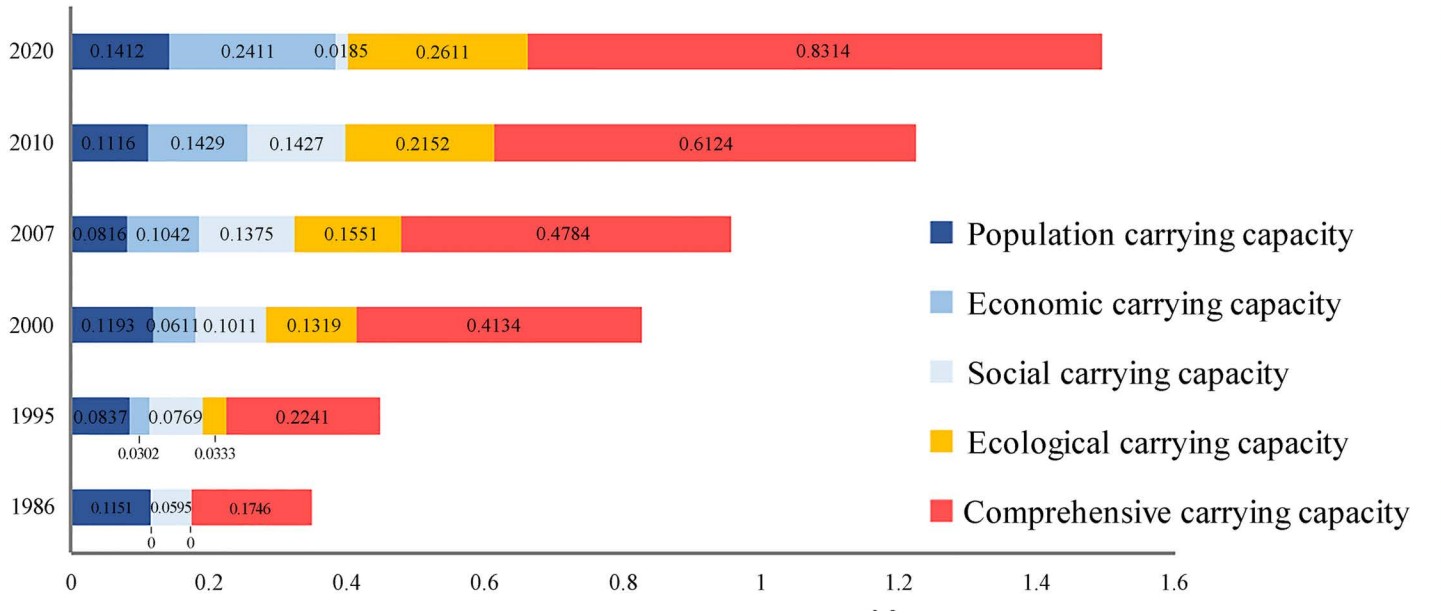

**Fig 9. Statistical figure of comprehensive evaluation values of land resource carrying ability in the TGRR.**

the evaluation values of the economic, social, and ecological carrying capacities of the land resources in the TGRR remained within the range of 0.1–0.15. Notably, the coordination between the various subsystems of land resource carrying ability at this stage significantly improved.

(5) The assessment value of the integrated carrying ability of land resources in 2010 in the TGRR was 0.6124, indicating that the level of land resource carrying ability reached a "basic" level. The ecological carrying ability of land resources at this stage was relatively good and showed an increasing trend. Simultaneously, the economic and social carrying capacity of land resources in the TGRR also improved to some degree, and its evaluation value remained in the range of 0.1–0.2, showing that the various carrying systems of land resources in the TGRR at this stage have been at a relatively balanced level.

(6) The assessment value of the integrated carrying ability of land resources in 2020 in the TGRR reached 0.8314, indicating that the land resource carrying ability reached a "high" level, approaching the optimal state value of comprehensive land resource carrying ability. During this period, the economic carrying ability of land resources in the TGRR increased the most, which also reflected the rapid economic growth of the TGRR. The evaluation values of the population and social carrying abilities of land resources continued to increase, further reflecting that the land structure of the TGRR remained relatively reasonable, and ecological environment protection and ecological civilization construction showed significant results.

## 4 Discussion and strategies

### 4.1 Discussion

Based on the panel data, such as remote sensing interpretation data and statistical yearbooks, and with the help of spatial information technologies, such as RS and GIS, and the introduction of research models, such as quantitative changes in the land use, land use dynamic indicators, and land use degree indicators, this study quantitatively analyzed the

spatiotemporal variation features of land use in the TGRR over the past 30 years. Based on the status quo of the research area and using the AHP evaluation method, a land resource carrying ability evaluation indicator model was constructed from the dimensions of social, economic, population, and ecological carrying abilities. A mean square error decision-making approach was introduced to determine the weight of the indicators, and the land resource carrying capacity of the TGRR from 1986 to 2020 was comprehensively analyzed. Overall, the research outcomes of this study were objective and credible.

(1) The methods used in this study may have certain limitations, mainly because of the subjectivity in the assessment of land resource carrying ability. For example, there may be differences in the research methods, perspectives, and evaluation indicators within the same research areas, resulting in different evaluation results. Nevertheless, the variation patterns of land resource carrying ability in the same region in the same year will be similar and will not affect the comparison of the research results [11,35–39].Upon reviewing the recommendations proposed in this study, we recognize that there is room for further improvement in the selection of data sources for future research. Specifically, while the current study has already provided a clear direction and suggestions for land resource management and sustainable development, in order to enhance the accuracy of land use classification and refine land use change detection, we believe that future research should consider adopting datasets with higher resolution. In this regard, satellite imagery such as Sentinel-2, with its high resolution and wide applicability, has become a highly potential data source. The multispectral band data provided by Sentinel-2 satellites can capture more subtle features of the earth's surface, thereby demonstrating higher accuracy in land use classification and change detection.

(2) Response of the land resource carrying ability to land use changes: As the most important water source and ecological conservation area in the Yangtze River Economic Belt, the TGRR underwent drastic variations in land use over the past 30 years [2,40,41]. Forestland was the major land use type that changed drastically, accounting for over 50% of the total area of the entire TGRR region. However, the forest land area showed a declining trend every year, reflecting a decrease of nearly 750 km$^2$ during the study period. Moreover, the construction land area in the TGRR showed the highest increase, from 234.59 km$^2$ in 1986 to 2061.87 km$^2$ in 2020, that is, a growth rate of 778.93%. Over the past 30 years, land use types in the research area have shifted from arable land and forest land to building land and water, which has also led to a synergistic change in the land resource carrying ability.

(3) Evaluation of land resource carrying ability has become a popular topic and a key direction in regional sustainable growth research. This study investigated the spatiotemporal features of land use variations in the TGRR from 1986 to 2020, and evaluated and analyzed its land resource carrying ability, which can serve as a reference for formulating land management policies and sustainable growth in the TGRR and other similar areas. However, some limitations exist. The data scale used in this study lacked absolute uniformity; moreover, errors may exist in the scale imbalance and uniformity while quantifying indicators. Additionally, collecting socioeconomic data in the TGRR was difficult, spatial scale of the data was large, and research on the spatial and local areas of the TGRR was lacking. Further investigations are needed to evaluate the carrying ability of regional land resources more accurately and to analyze the internal relationship and change mechanism between regional land resources and land use. Predicting the carrying capacity of land resources is an important problem. In future studies, we intend to further investigate these issues.

(4) Land use changes in the Three Gorges Reservoir Area exhibit significant regional differentiation patterns. A comparison with the Yangtze River Delta region shows that the intensity of construction land expansion in the reservoir area (average of 1.2% from 2010 to 2020) is less than one-third of the level in the Yangtze River Delta region (3.5%) during the same period, but the annual conversion rate of forest land (−0.8%) is four times that of the Yangtze River Delta region (−0.2%), revealing that the ecological barrier in the reservoir area faces stronger degradation

pressures. Compared with the Zhengzhou typical section (−4.3%), the reservoir area's farmland loss rate (−12.7%) is 8.4 percentage points higher, with large-scale resettlement projects confirmed as the core driving factor. International cases further highlight differences in development paths: based on the elevation gradient management experience of the Tennessee River Basin in the United States, this study proposes designating the 175–185 m elevation zone as the core area for intensive industrial development in the reservoir area; however, comparative data from the Itaipu Reservoir Area in Brazil show that the development intensity of tourism land in the Three Gorges Reservoir Area (5.1%) is only 27.9% of that in Itaipu (18.3%), indicating significant room for improvement in the ecological tourism industry [42–45].

(5) The primary reasons for the significant improvement in the ecological carrying capacity of the study area are as follows: The Grain-for-Green Program has played a foundational role in the reservoir area by increasing forest and grassland coverage, effectively controlling soil erosion, and enhancing water conservation capacity; the Natural Forest Resources Conservation Program is dedicated to protecting the integrity of existing forest ecosystems and maintaining biodiversity, holding significant long-term significance; the Three Gorges Reservoir Area Ecological Barrier Zone Construction and Reservoir Shore Zone Comprehensive Improvement Program focuses on stabilizing reservoir shores, managing the water level fluctuation zone, reducing sediment inflow into the reservoir, and ensuring water quality safety, achieving notable targeted results; and small watershed comprehensive management has played a positive role in controlling soil erosion at its source and improving local ecological environments. The coordinated advancement of these ecological projects has provided strong support for the restoration and sustainable development of the Three Gorges reservoir area's ecological system [46,47].

## 4.2 Strategies

Drawing upon the extensive research and analytical findings of our study, we conclude that optimizing the land carrying capacity of the TGRR and its Upstream Areas (TGRR) necessitates a holistic and multifaceted approach. This entails reinforcing land use planning and management frameworks to guarantee sustainable land utilization and striking a balance among the diverse needs of stakeholders, while embedding the fundamental principles of sustainability, ecological conservation, economic development, and now, green development.Our research highlights the paramount importance of intensifying ecological protection and restoration efforts through initiatives such as afforestation, wetland rehabilitation, and biodiversity preservation, which are vital for sustaining and enhancing the region's ecological resilience. Additionally, promoting green development strategies, such as the adoption of renewable energy sources, energy efficiency improvements, and the integration of green infrastructure into urban planning, will further strengthen the region's commitment to environmental stewardship. Moreover, by transforming the industrial structure towards knowledge-intensive and eco-friendly industries, fostering innovation, and advancing circular economy practices, we can bolster economic competitiveness without compromising environmental integrity. Integrating green technologies and sustainable practices across all economic sectors will ensure that growth is both environmentally sustainable and economically viable.Furthermore, our analysis underscores the necessity of robust population regulation and management strategies, encompassing the implementation of population control policies, the promotion of urbanization paired with rural revitalization, and the enhancement of public services. These measures, coupled with the promotion of green lifestyles and environmental education, are crucial for paving the way towards sustainable development. By meticulously balancing population growth, land use planning, economic development, and green development strategies, our study demonstrates that the TGRR can embark on a trajectory of harmonious and enduring progress. The strategies proposed herein, grounded in rigorous research and comprehensive analysis, are poised to make substantial contributions to the long-term sustainable development of the TGRR and its Upstream Areas, ensuring a future that is both prosperous and environmentally sustainable.

## 5  Conclusions and recommendations

### 5.1  Conclusions

This study used RS and GIS technologies to systematically analyze the spatiotemporal characteristics of quantitative changes in the land use type, speed of land use type changes, and land use degree in the TGRR from 1986 to 2020, using models related to land use variation. According to the indicator system of the "population–society–economy–ecology" dimension and using the carrying ability model, the land resource carrying ability was comprehensively evaluated, and the following conclusions were obtained:

(1) Between 1986 and 2020, land use types in the TGRR were mainly forest land and cropland. In particular, the area of built-up land and watersheds continued to increase, while the area of grassland, cropland and woodland decreased. In particular, the fastest increase was in construction land and the slowest change was in forest land. The composite degree index of land use types showed an increasing trend for all types of land use, indicating that the region was in a state of development during the study period, and that the fastest rate of land use development was observed during the period 1995–2000. In addition, the rate and process of change of different land use types showed diversity.

(2) From 1986 to 2020, the carrying capacity of land resources in the TGRR, including demographic, economic, social and ecological carrying capacity, generally increased, especially the ecological carrying capacity reached a high level. The comprehensive carrying capacity of land resources has gradually increased from weak to strong, and overall maintained a relatively stable upward trend without significant fluctuations. Especially during the period 1986–2010, the growth rate of the comprehensive carrying capacity was relatively slow, while in the subsequent period, the overall carrying capacity of land resources in the TGRR showed a rapid upward trend.

(3) During the study period, the land resources, demographic, economic, social and ecological factors of the TGRR have changed significantly, and the whole process of change has been more complicated. With the continuous increase in the area of construction land and watershed, and the decrease in the area of cultivated land, grassland and forest land, these changes have directly affected the carrying capacity of land. Specifically, this is manifested in the substantial increase in the carrying capacity of economic, social and ecological systems, especially the ecological carrying capacity is at a high level, occupying the largest proportion of the comprehensive carrying capacity of land resources in the TGRR. This indicates that the adjustment of land use structure is of great significance in enhancing the comprehensive carrying capacity of land and promoting sustainable development.

Overall, the results of this study are objective and authentic, and the construction of evaluation indicators can adapt to the regional and development characteristics of the TGRR. The evaluation method fully reflects the differences in land use changes and comprehensive carrying capacity development in different periods and regions of the TGRR, and reflects the contradictory relationship between land resources, ecological environment, and regional social and economic development. Finally, this study has certain reference significance and value in both the construction of evaluation indicators and the selection of evaluation methods in future land bearing capacity research.

### 5.2  Recommendations

In light of the ongoing changes in land resources within the Three Gorges Reservoir Region (TGRR), this study underscores the critical need for strengthened land use planning and regulatory oversight. The transformation of farmland and ecological land into construction land has been identified as a key factor constraining the region's social, economic, and ecological carrying capacities. Therefore, preserving high-quality arable land and essential ecological zones should be prioritized, while construction land allocation must be guided by principles of sustainability and spatial optimization.Moreover, the findings highlight that integrated land management strategies—such as promoting green development models, advancing technological innovation in land-use efficiency, and implementing adaptive governance—are instrumental in

enhancing regional resilience and long-term carrying capacity. To institutionalize these practices, it is imperative to refine existing policies and regulatory frameworks, ensuring they align with the ecological sensitivity and developmental needs of the TGRR.This study thus provides actionable insights for decision-makers, offering a science-based foundation for sustainable land resource management and policy formulation in ecologically vulnerable regions facing rapid land use transitions.

In terms of ecological protection, priority should be given to planting flood-tolerant vegetation in the 145–175m elevation zone and adopting stepped ecological slope protection technology to control soil erosion. Concurrently, the basin-wide collaborative governance mechanism should be refined, with the establishment of a Chongqing-Hubei interprovincial water quality monitoring alliance and the creation of a special fund for pollution compensation in the upstream reservoir area. In terms of green development, specific plans for industrial transformation should be outlined: Promote ecological agriculture through "citrus + traditional Chinese medicine" under-forest composite planting; prohibit the construction of new chemical industrial parks and relocate existing enterprises to circular economy transformation zones; implement a subsidy policy for ship LNG power conversion under green shipping initiatives (as outlined in the "Yangtze River Economic Belt Green Development Guidelines"); and implement technical application scenarios: construct distributed photovoltaic power stations in resettlement areas such as Zigui County, and utilize reservoir scheduling AI models to balance power generation with ecological flow. Additionally, new implementation safeguards are introduced, including the establishment of a green industry negative list for reservoir areas and the design of pathways for realizing the value of ecological products.

## Author contributions

**Data curation:** Xiaoyuan Zhang.

**Formal analysis:** Fuhai Wang, Honglei Guo.

**Investigation:** Yunmin Wang.

**Methodology:** Hui Li.

**Resources:** Yunmin Wang.

**Software:** Zhongshan Cui, Honglei Guo.

**Supervision:** Honglei Guo.

**Writing – original draft:** Hui Li.

**Writing – review & editing:** Hui Li.

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
