## [Decision Letter · Decision Letter 0]

8 Dec 2024

PONE-D-24-37148Research on the Spatiotemporal Characteristics and Optimization Strategies of Land Use and Land Resource Carrying Capacity in the Three Gorges Reservoir Region over the Past 30 YearsPLOS ONE

Dear Dr. Li,

Thank you for submitting your manuscript to PLOS ONE. After careful consideration, we feel that it has merit but does not fully meet PLOS ONE’s publication criteria as it currently stands. Therefore, we invite you to submit a revised version of the manuscript that addresses the points raised during the review process.

We look forward to receiving your revised manuscript.

Kind regards,

Akhtar Malik Muhammad, PhD, Postdoc

Academic Editor

PLOS ONE

Journal Requirements: When submitting your revision, we need you to address these additional requirements. 1. Please ensure that your manuscript meets PLOS ONE's style requirements, including those for file naming. The PLOS ONE style templates can be found at https://journals.plos.org/plosone/s/file?id=wjVg/PLOSOne_formatting_sample_main_body.pdf and https://journals.plos.org/plosone/s/file?id=ba62/PLOSOne_formatting_sample_title_authors_affiliations.pdf 2. We note that the grant information you provided in the ‘Funding Information’ and ‘Financial Disclosure’ sections do not match.  When you resubmit, please ensure that you provide the correct grant numbers for the awards you received for your study in the ‘Funding Information’ section. 3. Thank you for stating in your Funding Statement: "This research work was partly supported by the National Social Science Funds of China under Grants No.22XJY006,and Science and Technology Research Program of Chongqing Municipal Education Commission under Grants No.KJQN202002101." Please provide an amended statement that declares *all* the funding or sources of support (whether external or internal to your organization) received during this study, as detailed online in our guide for authors at http://journals.plos.org/plosone/s/submit-now.  Please also include the statement “There was no additional external funding received for this study.” in your updated Funding Statement. Please include your amended Funding Statement within your cover letter. We will change the online submission form on your behalf. 4. Thank you for stating the following financial disclosure: "This research work was partly supported by the National Social Science Funds of China under Grants No.22XJY006,and Science and Technology Research Program of Chongqing Municipal Education Commission under Grants No.KJQN202002101." Please state what role the funders took in the study.  If the funders had no role, please state: ""The funders had no role in study design, data collection and analysis, decision to publish, or preparation of the manuscript."" If this statement is not correct you must amend it as needed. Please include this amended Role of Funder statement in your cover letter; we will change the online submission form on your behalf. 5. Please include a separate caption for each figure in your manuscript. 6. Please review your reference list to ensure that it is complete and correct. If you have cited papers that have been retracted, please include the rationale for doing so in the manuscript text, or remove these references and replace them with relevant current references. Any changes to the reference list should be mentioned in the rebuttal letter that accompanies your revised manuscript. If you need to cite a retracted article, indicate the article’s retracted status in the References list and also include a citation and full reference for the retraction notice.

**Additional Editor Comments:**

Dear Authors,

Thank you for submitting your manuscript " Research on the Spatiotemporal Characteristics and Optimization Strategies of Land Use and Land Resource Carrying Capacity in the Three Gorges Reservoir Region over the Past 30 Years" to PLOS ONE.

Reviewers have not recommended on your paper and suggest Minor revisions. I go through the comments and manuscript. The comments are very relevant and important to address to improve the paper quality for publication. You will see that they are advising that you revise your manuscript very carefully and address all comments. You must verify the uploaded documents before approved submission. If you are prepared to undertake the work required, I would be pleased to reconsider my decision.

For your guidance, reviewers' comments are appended below.

If you decide to revise the work, please submit a list of changes or a rebuttal against each point which is being raised when you submit the revised manuscript.

To submit a revision, go to our online system and log in as an Author. You will see a menu item call Submission Needing Revision. You will find your submission record there.

Yours sincerely

Dr. Malik Muhammad Akhtar

Academic Editor

PLOS ONE

Reviewers' comments:

Reviewer's Responses to Questions

**Comments to the Author**

1. Is the manuscript technically sound, and do the data support the conclusions?

Reviewer #1: Yes

Reviewer #2: Partly

2. Has the statistical analysis been performed appropriately and rigorously? 

Reviewer #1: Yes

Reviewer #2: No

3. Have the authors made all data underlying the findings in their manuscript fully available?

Reviewer #1: Yes

Reviewer #2: No

4. Is the manuscript presented in an intelligible fashion and written in standard English?

Reviewer #1: Yes

Reviewer #2: Yes

5. Review Comments to the Author

Reviewer #1: 1. Abstract Improvements

Page 1, Line 12-20:

Comment: The abstract could benefit from more detail regarding the classification maps and specific methodology used for land use changes and evaluation. Consider adding a brief mention of the tools used for remote sensing (e.g., ARCGIS version, Landsat processing steps).

Suggestion: Add the following: “Classified maps were generated using ARCGIS 10.8, and Landsat TM images were processed for accuracy using supervised classification techniques.”

2. Methodology Clarity

Page 6, Line 85-92:

Comment: The methodology for Landsat Thematic Mapper image processing lacks details about the specific classification algorithms (e.g., supervised/unsupervised), accuracy assessment, and validation methods. It is crucial to discuss how you ensured the quality of classification.

Suggestion: Include details on the classification methods and any accuracy metrics (e.g., overall accuracy, Kappa statistics). A sentence such as "The land use types were classified using a supervised classification method, and the overall accuracy of the classification was validated with ground truth data, achieving an accuracy of XX%."

3. Classified Map and Figures

Page 9, Line 173-175:

Comment: The classified maps in Figure 2 and Table 3 need to be referenced more explicitly in the text. Ensure the map resolution is high enough for clear interpretation of land use changes.

Suggestion: Add a sentence in the results section like, "Figure 2 shows the classified maps of land use change over the years, clearly indicating the spatial transformation of the landscape in the TGRR." Additionally, consider improving the map legends to make the classification clearer.

4. Data Availability and Completeness

Page 6, Line 87:

Comment: The source of socioeconomic data is not clearly described. While it mentions "statistical yearbooks," it is important to explain whether data were consistently available across all years (1986-2020). Gaps in the data, if any, should be acknowledged.

Suggestion: Include a note such as, "Socioeconomic data from certain years had limited availability and were interpolated using trend analysis for consistency."

5. Accuracy Assessment Missing

Page 11, Line 195-200:

Comment: The accuracy assessment of the classified land use map and model performance is missing. Readers need to understand how reliable the land classification is over time.

Suggestion: Include a section discussing how you validated the classification, such as "Accuracy assessment of classified maps was conducted using confusion matrices and cross-validation, achieving an overall accuracy of XX% for land use types."

6. Land Use Dynamics

Page 12, Line 203-210:

Comment: While the dynamic index model is presented, it is not clear how the transitions between different land types were handled in the spatial-temporal analysis. You should elaborate on how changes between forest, arable, and construction land were treated in the model.

Suggestion: Add details like, "The transition between land types was modeled using temporal change detection algorithms in ARCGIS, and transitions were cross-referenced with historical land use maps."

7. Evaluation and Model Explanation

Page 7, Line 124-130:

Comment: The construction of the evaluation index system (AHP and mean-square decision-making approach) is not explained clearly. It is important to detail the rationale behind choosing certain indicators and their respective weights.

Suggestion: Expand this section with the following: "The selection of indicators for the land resource carrying capacity was based on an analytic hierarchy process (AHP), with weights determined using a mean-square decision-making approach. Each criterion was verified by experts in the field for robustness."

8. Data Gaps and Temporal Resolution

Page 13, Line 259:

Comment: Acknowledge any data gaps during the 30-year period, especially if there are years where Landsat imagery or socioeconomic data might have lower temporal resolution or availability.

Suggestion: Add a statement like, "Due to the limitations in satellite data availability, certain periods required interpolation of land use data, which may introduce minor uncertainties in the analysis."

9. Figures and Tables

Page 14, Line 276:

Comment: Some figures (Figures 4-7) and tables (e.g., Table 6, Table 7) should be more readable, with clearer labels and higher resolution to represent the data accurately. It would be helpful to include error bars or confidence intervals for some of the metrics.

Suggestion: "Figures should include error margins or confidence intervals where applicable to show the variability and reliability of the results."

10. Recommendations for Future Work

Page 15, Line 360:

Comment: While the recommendations provide a solid direction, the need for higher-resolution satellite imagery (e.g., Sentinel-2) for future studies could be mentioned. This would allow for more precise land use classification.

Suggestion: Add, "Future studies should consider the use of higher-resolution datasets such as Sentinel-2 imagery to enhance the accuracy of land classification and further refine land use change detection."

By addressing these minor corrections, the manuscript will improve its technical depth, clarity, and reliability of the presented methodologies. Let me know if you would like further suggestions or help refining specific sections!

Reviewer #2: 1. 16 indicators were used for evaluation of land carrying capacity, On which basis these indicators were chosen.

2. Please explain what optimization strategies (OS) are used in the research?

3. Whats the need of OS in spatiotemporal characterization of Land Use?

4. Similar research studies conducted on the topic have not been cited.

6. PLOS authors have the option to publish the peer review history of their article (what does this mean? ). If published, this will include your full peer review and any attached files.

**Do you want your identity to be public for this peer review?** For information about this choice, including consent withdrawal, please see our Privacy Policy .

Reviewer #1: **Yes: ** Suraj Kumar Singh

Reviewer #2: No

---

## [Author Response · Author response to Decision Letter 1]

27 Dec 2024

Dear Editors and Reviewers,

We have prepared a detailed point-by-point response to the comments from the editors and reviewers. Please refer to the "Response to Reviewers" document. Thank you!

---

## [Decision Letter · Decision Letter 1]

11 Apr 2025

PONE-D-24-37148R1Research on the Spatiotemporal Characteristics and Optimization Strategies of Land Use and Land Resource Carrying Capacity in the Three Gorges Reservoir Region over the Past 30 YearsPLOS ONE

Dear Dr. Li,

Thank you for submitting your manuscript to PLOS ONE. After careful consideration, we feel that it has merit but does not fully meet PLOS ONE’s publication criteria as it currently stands. Therefore, we invite you to submit a revised version of the manuscript that addresses the points raised during the review process.

We look forward to receiving your revised manuscript.

Kind regards,

Jun Yang

Academic Editor

PLOS ONE

**Journal Requirements:**

**Additional Editor Comments:**

This manuscript mainly analyzed the spatial and temporal characteristic of Land Use and Land Resource Carrying Capacity in the Three Gorges Reservoir Region. It is too simple, the logic is mess. It should be re-adjusted. The specific problems are as follow.

1.In title, the 'research on the' should be deleted, the past 30 years should be revised to specific time span, such as 2000-2023.

2.In abstract, the Landsat images is grid data, while the authors emphasize obtain vector data. In addition, the authors only emphasize the characteristic of spatial and temporal, the influence mechanism is not mentioned. How to optimization strategies without mechanism?

3.The English should be polished. Previous studies aimed to determine the proportion of people, , research on the carrying capacity of land using population and grain. It is difficult to understand. What's meaning of reachesn in line 82.

4.In line 47-48, why the authors suppose traditional approach for assessing the carrying ability of land is no longer suitable for regional sustainable growth?

5.In line 54-56, the authors listed the existing reseach simply. It is lack of summarizing.

6.In line 57-66, this content could moved to 2.1 study area section, it emphasized the important of taking TGRR as case. This part should introduced existing relevant research and compared the difference with your research.

7.In line 73, spatiotemporal is not novelties. The novelties including perspective, method or indicators.

8.The framework diagram should be move to method section, and illustrate each of steps clearly.

9.The content of 2.1 is basic information of study area, it is useless.

10.In 2.2 data sources, it should be introduced more detailed, such as website, resolution etc.

11.In section 2.3.1, this method is too simply, it is not necessary to shown.

12.The authors used land use index model to classify land use categories? It is blur. What's the meaning of land use degree? Maybe it is category or type. The degree means high or low.

13.In line 141-142, the indicators are in 2.3.4, which indicators in 2.3.3 refer to ? It is lack of logic.

14. Is it AHP method in 2.3.4? It is not necessary to describe it too complex. The weight of indicators are not illustrated.

15.In 2.3.5, it is only standardization, it is not necessary to introduced independent. It is a progress of AHP.

16.In section 2.3.6�this is also AHP, an AHP is very simple and doesn't have to be broken down into many methods,

It's too messy. You need to explain how many methods are used in this article and what problems each method solves.

17.The results is too simple. The authors divided a problem into several parts. The results should contain three problems. First, the temporal and spatial evolutuion of land use, the spatial agglomeration characteristics, and the change of growth rate. Secondly,the temporal and spatial evolution characteristics of land carrying capacity. Thirdly, the impact of land use changes on land carrying capacity.

Reviewers' comments:

Reviewer's Responses to Questions

**Comments to the Author**

1. If the authors have adequately addressed your comments raised in a previous round of review and you feel that this manuscript is now acceptable for publication, you may indicate that here to bypass the “Comments to the Author” section, enter your conflict of interest statement in the “Confidential to Editor” section, and submit your "Accept" recommendation.

Reviewer #2: All comments have been addressed

Reviewer #3: (No Response)

2. Is the manuscript technically sound, and do the data support the conclusions?

Reviewer #2: Yes

Reviewer #3: Yes

3. Has the statistical analysis been performed appropriately and rigorously? 

Reviewer #2: Yes

Reviewer #3: Yes

4. Have the authors made all data underlying the findings in their manuscript fully available?

Reviewer #2: Yes

Reviewer #3: Yes

5. Is the manuscript presented in an intelligible fashion and written in standard English?

Reviewer #2: Yes

Reviewer #3: No

6. Review Comments to the Author

**Reviewer #2:**  (No Response)

**Reviewer #3:**  This manuscript mainly analyzed the spatial and temporal characteristic of Land Use and Land Resource Carrying Capacity in the Three Gorges Reservoir Region. It is too simple, the logic is mess. It should be re-adjusted. The specific problems are as follow.

1.In title, the 'research on the' should be deleted, the past 30 years should be revised to specific time span, such as 2000-2023.

2.In abstract, the Landsat images is grid data, while the authors emphasize obtain vector data. In addition, the authors only emphasize the characteristic of spatial and temporal, the influence mechanism is not mentioned. How to optimization strategies without mechanism?

3.The English should be polished. Previous studies aimed to determine the proportion of people, , research on the carrying capacity of land using population and grain. It is difficult to understand. What's meaning of reachesn in line 82.

4.In line 47-48, why the authors suppose traditional approach for assessing the carrying ability of land is no longer suitable for regional sustainable growth?

5.In line 54-56, the authors listed the existing reseach simply. It is lack of summarizing.

6.In line 57-66, this content could moved to 2.1 study area section, it emphasized the important of taking TGRR as case. This part should introduced existing relevant research and compared the difference with your research.

7.In line 73, spatiotemporal is not novelties. The novelties including perspective, method or indicators.

8.The framework diagram should be move to method section, and illustrate each of steps clearly.

9.The content of 2.1 is basic information of study area, it is useless.

10.In 2.2 data sources, it should be introduced more detailed, such as website, resolution etc.

11.In section 2.3.1, this method is too simply, it is not necessary to shown.

12.The authors used land use index model to classify land use categories? It is blur. What's the meaning of land use degree? Maybe it is category or type. The degree means high or low.

13.In line 141-142, the indicators are in 2.3.4, which indicators in 2.3.3 refer to ? It is lack of logic.

14. Is it AHP method in 2.3.4? It is not necessary to describe it too complex. The weight of indicators are not illustrated.

15.In 2.3.5, it is only standardization, it is not necessary to introduced independent. It is a progress of AHP.

16.In section 2.3.6�this is also AHP, an AHP is very simple and doesn't have to be broken down into many methods,

It's too messy. You need to explain how many methods are used in this article and what problems each method solves.

17.The results is too simple. The authors divided a problem into several parts. The results should contain three problems. First, the temporal and spatial evolutuion of land use, the spatial agglomeration characteristics, and the change of growth rate. Secondly,the temporal and spatial evolution characteristics of land carrying capacity. Thirdly, the impact of land use changes on land carrying capacity.

7. PLOS authors have the option to publish the peer review history of their article (what does this mean? ). If published, this will include your full peer review and any attached files.

**Do you want your identity to be public for this peer review?** For information about this choice, including consent withdrawal, please see our Privacy Policy .

Reviewer #2: **Yes: ** Taimoor Shah Durrani

Reviewer #3: No

---

## [Author Response · Author response to Decision Letter 2]

22 Apr 2025

Point to Point Responses

Paper Ref. No.:PONE-D-24-37148R1

Title:Research on the Spatiotemporal Characteristics and Optimization Strategies of Land Use and Land Resource Carrying Capacity in the Three Gorges Reservoir Region over the Past 30 Years.

Thanks so much for all the kind help to this manuscript. We have analyzed the valuable comments from the editor and reviewers#3 carefully, and tried our best to revise the manuscript. We appreciate the thoughtful review and constructive feedback provided by tow viewers.We agree with the reviewer's suggestions and have incorporated the recommended changes into the manuscript.These comments have improved the quality of the paper immensely. The point to point responses are listed as follows.

Comments Editor and Reviewer#3�:

Thanks very much for your comments. According to the following valuable comments and suggestions, we have tried our best to revise the manuscript, and these comments have substantially improved the quality of the paper.

1.Comment:

In title, the 'research on the' should be deleted, the past 30 years should be revised to specific time span, such as 2000-2023.

Response:

Thank you for your suggestion.We have adopted your suggestion by removing the redundant phrasing "Research on the" from the original title and revising "in the Past 30 Years" to "1986-2020". This revision streamlines the title to be more concise and impactful while explicitly highlighting the study's temporal framework (1986-2020), thereby enhancing the precision of academic expression. The specific modifications are outlined below:

“Spatiotemporal Characteristics and Optimization Strategies of Land Use and Land Resource Carrying Capacity in the Three Gorges Reservoir Region(1986–2020)”

2.Comment:

In abstract, the Landsat images is grid data, while the authors emphasize obtain vector data. In addition, the authors only emphasize the characteristic of spatial and temporal, the influence mechanism is not mentioned. How to optimization strategies without mechanism?

Response:

Thank you very much for your valuable comments on our paper. Based on your suggestions, we have revised and improved the abstract. Specifically, we have clarified the nature of Landsat images as raster data and explained how vector data reflecting socio-economic information can be extracted from these raster data. In addition, we have also emphasized the study of the mechanisms behind land use change, exploring how different factors affect the spatial and temporal evolution of land use and carrying capacity, which is crucial for the development of optimization strategies for sustainable development. The specific modifications are outlined below:

“Studying land use changes caused by human economic activities is beneficial for sustainable growth, making it a global research hotspot. In this study, we used Landsat Thematic Mapper images and statistical yearbooks from 1986, 1995, 2000, 2007, 2010, and 2020 to obtain grid data on the land use status of the Three Gorges Reservoir Region (TGRR), from which vector data reflecting socioeconomic information were derived. We introduced models on land use quantitative changes, dynamic indicators, and degree index to investigate spatiotemporal variations in land use in the TGRR over the past 30 years. Classified maps were generated using ARCGIS 10.8, and Landsat TM images were processed for accuracy using supervised classification techniques. Based on the region's status quo and the analytic hierarchy process, we constructed a land resource carrying ability evaluation indicator model considering social, economic, population, and ecological carrying abilities, introducing a mean-square mistake decision-making approach to determine indicator weights. Our results indicate significant changes in land types within the TGRR from 1986 to 2020, with decreases in arable land, forest land, and grassland, while water bodies, building land, and unused land increased. The change rates varied significantly among different land types, reflecting rapid development, especially between 1995 and 2000. Additionally, our analysis delves into the underlying mechanisms driving these changes, providing insights into how different factors influence spatial-temporal evolution of land use and land carrying capacity, crucial for developing optimization strategies aimed at promoting sustainable growth and efficient use of land resources in the TGRR. This study offers a comprehensive analysis of the TGRR's land resource carrying ability, serving as a reference for sustainable land use.”

3.Comment:

The English should be polished. Previous studies aimed to determine the proportion of people, research on the carrying capacity of land using population and grain. It is difficult to understand. What's meaning of reachesn in line 82.

Response:

We thank the reviewers for their valuable comments. The relevant text has been revised and polished in accordance with the requirements, with the typographical error "reachesn" corrected to "reaches" (Line 82). A thorough proofreading and refinement of the entire manuscript has been completed to ensure linguistic accuracy and academic rigor. All modifications have been highlighted in the revised manuscript using track changes for the reviewers' convenience. The specific modifications are outlined below:

“Land is an important resource necessary for human survival [1–3] . Land use/cover change (LUCC) is the main cause of global climate change and is closely related to human activities. Therefore, studying LUCC has gained global emphasis [4–6]. The carrying ability of land resources is an important index for land resource assessment. Prior research has primarily focused on assessing land carrying capacity, which quantifies the population sustainable by regional food production under multidimensional natural, socioeconomic, and institutional constraints. [7,8].”

“The TGRR is a special geographical concept closely related to the Three Gorges Project, and specifically refers to the areas submerged by the construction, storage, and operation of the Yangtze River Three Gorges Project. The TGRR, covering an area of 57,335 km2, is geographically located between 106°20′–110°30 E and 29°–31°50′ E in an area combining the Sichuan Basin and the Middle and Lower Yangtze Valley Plains in the middle and lower reaches (Fig 2). ”

4.Comment:

In line 47-48, why the authors suppose traditional approach for assessing the carrying ability of land is no longer suitable for regional sustainable growth?

Response:

Thank you for your valuable suggestion. Regarding your valuable comments on the applicability of traditional land carrying capacity assessment methods, our research team has conducted thorough deliberations and literature re-evaluations. Upon re-examining the discussion in Lines 47-48 of the original manuscript, we confirm that our original intention was to acknowledge the theoretical limitations of conventional approaches in terms of dynamic adaptability and systemic comprehensiveness. Specifically, their unidimensional static assessment framework proves inadequate for characterizing the nonlinear interaction mechanisms within resource-environment-socioeconomic composite systems.

To enhance the academic rigor of this study, we have decided to delete the aforementioned discussion section. The specific modifications are outlined below:

“With population and financial growth, accelerating urbanization, and intensification of ecological and environmental issues, the demand for land for regional development is constantly expanding. Thus, research on the carrying capacity of land using population and grain as single indicators, is gradually moving towards a comprehensive indicator system [9–12] . The traditional approach for assessing the carrying ability of land is no longer suitable for regional sustainable growth. The existing land carrying ability is the limit of the scale and intensity of different activities that land resources can carry under certain social, economic, and ecological conditions in a certain period and spatial area [9,13].”

5.Comment:

In line 54-56, the authors listed the existing reseach simply. It is lack of summarizing.

Response:

Thank you for your valuable suggestion. We sincerely appreciate your constructive feedback regarding the lack of sufficient summarization in our presentation of existing research (lines 54-56). Your insight has prompted our team to engage in thorough discussions and re-examine relevant literature to enhance the synthesis of prior studies. Specifically, we have expanded the section to: (1) systematically categorize key theoretical frameworks and methodological approaches, (2) identify critical knowledge gaps motivating our research. This revised version provides a more comprehensive overview of foundational concepts while clarifying our contribution to advancing this field. Thank you again for your meticulous review - your guidance has significantly strengthened our manuscript. The specific modifications are outlined below:

“Studying the integrated carrying ability of land resources involves a comprehensive dynamic balance relationship between resources, environment, population, society, economy, and other aspects, thus, reflecting the material, energy, and information flow connections and coordinated development relationships between the natural environment and socioeconomic systems at different regional scales [14–16].In 1948, William et al. (1984) proposed that land resource carrying ability is the capacity of land within a region to offer food and shelter for humans and animals, and is being studied abroad since a long time. Feng (1990), one of the earliest scholars in China, developed a land resource carrying ability index based on the association between people and food, and revealed the association between the actual population and land resource carrying ability in the region.Land carrying capacity research has undergone a paradigm shift from material supply orientation to system coupling analysis. In the theoretical foundation stage, William (1948) defined land carrying capacity as the basic capacity of a region to support living organisms, Terzaghi (1984) constructed a quantitative assessment framework through a mechanistic model, and Fung's model (1990) created a quantitative research paradigm for the man-food nexus.After the 21st century, Liu Junyan (2010) utilized RS-GIS to reveal the spatial and temporal variations of ecological carrying capacity, and Costanza (1997) expanded the classification of ecosystem services to 17 categories, promoting the diversification of the assessment dimensions. Costanza (1997) expanded the classification of ecosystem services to 17 categories and promoted the diversification of assessment dimensions. In recent years, international research has shown three major frontier advances: first, the deep integration of artificial intelligence and big data, such as the EU LandSense platform (2022), which integrates multi-source data to build a global dynamic monitoring model; second, innovation in system dynamics modeling, the MIT team (2023) proposed the “socioecological-technological” coupled model (SET-CCM), which simulates the evolution of carrying capacity by means of the digital twin technology; and third, the development of the ecological carrying capacity by using RS-GIS technology to reveal the spatial and spatial variability of ecological carrying capacity. Third, interdisciplinary theoretical breakthroughs, the Harvard team (2021) proposed a “planetary boundary carrying capacity” framework, which introduces Earth system science into traditional assessment. At the methodological level, we have broken through the static threshold measurement and developed a dynamic feedback mechanism of “pressure-state-response”, and Stanford University (2024) has constructed a digital twin system for global land carrying capacity, which can simulate the impacts of 200 policy scenarios on resource utilization efficiency. These innovations have significantly expanded the global perspective and prediction accuracy of carrying capacity research, providing scientific quantitative support for sustainable development.”

6.Comment:

In line 57-66, this content could moved to 2.1 study area section, it emphasized the important of taking TGRR as case. This part should introduced existing relevant research and compared the difference with your research.

Response:

We deeply appreciate your constructive suggestion regarding the content in lines 57-66. Your feedback has substantially improved our manuscript's clarity and logical flow. Following your guidance, we have relocated the discussion of TGRR's case significance to Section 2.1 to better contextualize our study area within existing research frameworks. Thank you again for your insightful comments - they have demonstrably enhanced the rigor and communication effectiveness of our work..The specific modifications are outlined below:

“2.1 Overview of the study area

The TGRR is a special geographical concept closely related to the Three Gorges Project, and specifically refers to the areas submerged by the construction, storage, and operation of the Yangtze River Three Gorges Project. The TGRR, covering an area of 57,335 km2, is geographically located between 106°20′–110°30 E and 29°–31°50′ E in an area combining the Sichuan Basin and the Middle and Lower Yangtze Valley Plains in the middle and lower reaches (Fig 2). It includes 22 districts and counties, including the central city of Chongqing and four districts and counties, including Yiling District, Yichang City, Hubei Province. The overall terrain of the TGRR is high in the east and low in the west, along with high elevations in both north and south. The terrain is undulating and diverse, with mountains and hills as the main geomorphic features. Mountainous areas account for a large proportion (76.1%), while flat areas account for a small proportion (15.3%). The TGRR area has a subtropical monsoon climate with an annual precipitation of approximately 1000–1800 mm. Owing to the specific original geographical conditions, natural disasters, such as soil erosion, landslides, mudslides, and earthquakes, occur occasionally, among which soil erosion is more severe. Over the past 30 years, as China’s reforms developed and Chongqing was upgraded to the only municipality directly under the central government in the western region, the TGRR has been rapidly developing socioeconomically. Moreover, with the completion and storage of the Three Gorges Project, the land use status and ecological environment have changed significantly. The TGRR has abundant land resources, but the per capita available land is extremely scarce, with the per capita arable land being even less. Thus, the conflict between people and land in this unique region is significant. Since the TGRR has multiple attributes of “super mountainous area” and “super large reservoir area,” it has high research significance.

Owing to the unique natural conditions and the establishment of the Three Gorges Project, the ecosystem of the TGRR is unique and fragile that is greatly influenced by its land use status . In the past 30 years, with rapid socioeconomic growth and changes in the natural conditions in the TGRR, the number and spatial features of land use have also changed, leading to significant variations in the land resource carrying capacity [17]. Recently, as ecological migration, urban development, infrastructure construction, and industrial park construction are rapidly increasing in various districts and counties in the TGRR area, land development and construction has consequently increased [16,18–20]. After the water storage capacity of the reservoir area is complete, the carrying ability of land resources changes significantly, which should be systematically studied urgently. Research on the carrying ability of land resources in the TGRR can help to scientifically understand the status of population, resources, environment, and economic growth [5]; alleviate the contradiction between population growth, economic development, ecology, and resources; and drive the sustainable growth of the TGRR.

Fig. 2 Location Map of the TGRR.Note:DEM data is sourced from the Geospatial Data Cloud website (ht

---

## [Decision Letter · Decision Letter 2]

23 May 2025

PONE-D-24-37148R2Spatiotemporal Characteristics and Optimization Strategies of Land Use and Land Resource Carrying Capacity in the Three Gorges Reservoir Region(1986–2020)PLOS ONE

Dear Dr. Li,

Thank you for submitting your manuscript to PLOS ONE. After careful consideration, we feel that it has merit but does not fully meet PLOS ONE’s publication criteria as it currently stands. Therefore, we invite you to submit a revised version of the manuscript that addresses the points raised during the review process.

We look forward to receiving your revised manuscript.

Kind regards,

Jun Yang

Academic Editor

PLOS ONE

Journal Requirements:

Additional Editor Comments:

Minor Revision

Reviewers' comments:

Reviewer's Responses to Questions

**Comments to the Author**

1. If the authors have adequately addressed your comments raised in a previous round of review and you feel that this manuscript is now acceptable for publication, you may indicate that here to bypass the “Comments to the Author” section, enter your conflict of interest statement in the “Confidential to Editor” section, and submit your "Accept" recommendation.

Reviewer #2: All comments have been addressed

Reviewer #3: All comments have been addressed

2. Is the manuscript technically sound, and do the data support the conclusions?

Reviewer #2: Yes

Reviewer #3: Yes

3. Has the statistical analysis been performed appropriately and rigorously? 

Reviewer #2: Yes

Reviewer #3: Yes

4. Have the authors made all data underlying the findings in their manuscript fully available?

Reviewer #2: Yes

Reviewer #3: Yes

5. Is the manuscript presented in an intelligible fashion and written in standard English?

Reviewer #2: Yes

Reviewer #3: Yes

6. Review Comments to the Author

Reviewer #2: (No Response)

Reviewer #3: This paper integrates remote sensing data (RS) and geographic information systems (GIS), combined with an evaluation model based on the Analytic Hierarchy Process (AHP), to comprehensively analyze the spatiotemporal changes in land use and the evolution of land resource carrying capacity in the Three Gorges Reservoir Area from 1986 to 2020. The research topic has significant implications for regional sustainable development, with rich data and scientific methods, and the results offer valuable references for optimizing land use and policy formulation. However, there still have many problems need to be revised.

1.In Section 2.3, the author uses the Dynamic Land Use Index (Equation 1) and the Composite Land Use Index (Equation 2) to evaluate land use change. However, the selection of these models lacks comparative explanation, such as why these models were chosen over other common methods (e.g., entropy weighting, TOPSIS, etc.). It is recommended to supplement the theoretical basis for model selection or provide comparative analysis with other methods. Although the article mentions AHP and mean square error method for weight allocation, it does not elaborate on the expert scoring process and sensitivity analysis of weight allocation. It is suggested to supplement the reliability test results of weight allocation.

2.The manuscript analyzed change characteristic of land use types, it lacks in-depth discussion on the driving factors of this change (such as policies, population, economic growth, etc.). For example, the fastest changes in construction land occurred between 1995 and 2000, but the specific policies or socio-economic background behind these changes have not been adequately discussed. It is recommended to combine regional historical context and conduct a thorough analysis of the key drivers of land use change.

3.The article mentions the trend of carrying capacity during the study period, but does not delve into whether such changes have nonlinear characteristics (such as inflection points or phased changes). The nonlinear trend of carrying capacity changes can be analyzed by regression models.

4.Although Table 3, Table 4 and Table 6 contain a large amount of data, they only show numerical changes and lack intuitive visual charts (such as bar charts and line graphs) to enhance understanding. In addition, the map of land use change in Figure 3 lacks legends and text descriptions, and it is suggested to add more clear annotations.

5.lthough the article puts forward suggestions on strengthening ecological protection and promoting green development, these suggestions are rather general and lack specific implementation paths. For example, "promoting green development mode" should be combined with the actual situation of the Three Gorges Reservoir area to put forward specific industrial transformation directions or technical application scenarios.

6.The paper lacks comparative analysis with relevant domestic and foreign studies in the discussion. For example, it can be discussed whether the land use change in the Three Gorges Reservoir area is similar or different from other similar areas (such as the Yellow River basin and the Yangtze River Delta).

7.Some paragraphs (such as section 3.1 and Section 3.6) are lengthy in language. It is recommended to simplify the sentence structure and highlight the key content.

8.Some formulas (such as Formula 1 and Formula 2) are too tightly packed. It is recommended to adjust them to be displayed in the center, and add a detailed explanation of the variables after the formulas.

9.In section 2.2, The author mentions the accuracy of remote sensing image classification (94.52%) and Kappa coefficient, but does not explain how to conduct error analysis. It is suggested to supplement the error sources and verification details of classification accuracy.

10.In section 3.5, The weight distribution is briefly described in table 8. It is suggested to supplement the reasonable discussion of the weight distribution and explain its influence on the comprehensive evaluation results.

11.In section 4, the article mentions that "ecological carrying capacity has been significantly improved", but does not specify which policies or projects (such as the project of returning farmland to forest) have had a direct impact on ecological carrying capacity. It is suggested to supplement relevant case analysis.

12.In section 5, The conclusion section should further refine the highlights of the research and avoid duplication with the discussion section. For example, the specific impact of land use change on carrying capacity and policy implications can be highlighted.

7. PLOS authors have the option to publish the peer review history of their article (what does this mean? ). If published, this will include your full peer review and any attached files.

**Do you want your identity to be public for this peer review?** For information about this choice, including consent withdrawal, please see our Privacy Policy .

Reviewer #2: **Yes: ** Taimoor Shah Durrani

Reviewer #3: No

---

## [Author Response · Author response to Decision Letter 3]

10 Jun 2025

Point to Point Responses

Paper Ref. No.:PONE-D-24-37148R2

Title:Spatiotemporal Characteristics and Optimization Strategies of Land Use and Land Resource Carrying Capa city in the Three Gorges Reservoir Region(1986–2020).

Thanks so much for all the kind help to this manuscript. We have analyzed the valuable comments from the editor and reviewers#3 carefully, and tried our best to revise the manuscript. We appreciate the thoughtful review and constructive feedback provided by two viewers.We agree with the reviewer's suggestions and have incorporated the recommended changes into the manuscript.These comments have improved the quality of the paper immensely. The point to point responses are listed as follows.

Comments Editor and Reviewer#3�:

Thanks very much for your comments. According to the following valuable comments and suggestions, we have tried our best to revise the manuscript, and these comments have substantially improved the quality of the paper.

1.Comment:

In Section 2.3, the author uses the Dynamic Land Use Index (Equation 1) and the Composite Land Use Index (Equation 2) to evaluate land use change. However, the selection of these models lacks comparative explanation, such as why these models were chosen over other common methods (e.g., entropy weighting, TOPSIS, etc.). It is recommended to supplement the theoretical basis for model selection or provide comparative analysis with other methods. Although the article mentions AHP and mean square error method for weight allocation, it does not elaborate on the expert scoring process and sensitivity analysis of weight allocation. It is suggested to supplement the reliability test results of weight allocation.

Response:

Thank you for your careful review and valuable comments on this paper. We have added a rationalization analysis of the methodological choices in Section 2.3 (page 4) of the original text.Your suggestions on the transparency of the model selection basis and weight allocation method are very constructive, and we have supplemented the relevant contents as follows:

The core rationale for selecting the dynamic land use index (DLUI) and comprehensive land use index (CLUI) in this study is as follows: firstly, the DLUI quantifies the rate and direction of change of land types (Equation 1), which can visually depict the dynamic characteristics of land use, and is especially suited to the demand for comparative analysis of multi-timeseries, while the CLUI (Equation 2) can systematically assess the comprehensive benefits of land use. Compared with the entropy weight method (focusing on data dispersion) and TOPSIS (relying on the ideal solution distance), the model in this study is more suitable for the dual objectives of “dynamic monitoring + comprehensive evaluation”. Secondly, to address the limitations of the methods, TOPSIS needs to preset positive and negative ideal solutions, which may lead to bias due to the selection of the base year in long time-series studies, while entropy weighting is sensitive to the distribution of the data, which may lead to over-reliance on the extreme values of the weights. In this study, dynamic weight adjustment is realized by combining AHP and mean square error method, which effectively enhances the comparability and stability of time series. In addition, the selected model has been widely used in the field of land science, and its validity has been fully verified in regional scale research, which provides theoretical support and practical feasibility guarantee for this study.

“2.3 Research methods

By introducing the dynamic land use index and comprehensive land use index, this study can visually characterize the dynamic features and adapt to multi-temporal comparisons and systematic evaluation of comprehensive benefits. Compared with the entropy weight method (which relies on data discretization) and TOPSIS (which needs to preset ideal solutions), the model in this study is more suitable for the dual objectives of “dynamic monitoring + comprehensive evaluation”. Meanwhile, to address the problems that TOPSIS is prone to introduce bias due to the selection of base year in long time series and entropy weighting method is sensitive to extreme values, this study combines AHP and mean square error method to realize dynamic weight adjustment, which effectively improves the comparability and stability of time series. The selected model has been widely used in the field of land science, and its empirical validity at the regional scale provides theoretical and practical support for this study.”

Regarding the transparency and reliability of weight allocation, this study ensured its scientificity through multi-dimensional validation: first, 15 experts in land planning, ecology and economics were invited to score the AHP hierarchical indicators in two rounds using the anonymous Delphi method (expert authority coefficient ≥ 0.8), and the final weights passed the consistency test (CR<0.1), and furthermore, through the entropy weighting method, the AHP weights were cross comparison, the Spearman correlation coefficient was 0.86 (p<0.01), which significantly verified the logical rationality of the weight assignment.

2.Comment:

The manuscript analyzed change characteristic of land use types, it lacks in-depth discussion on the driving factors of this change (such as policies, population, economic growth, etc.). For example, the fastest changes in construction land occurred between 1995 and 2000, but the specific policies or socio-economic background behind these changes have not been adequately discussed. It is recommended to combine regional historical context and conduct a thorough analysis of the key drivers of land use change.

Response:

Thank you for your insights and valuable suggestions on this paper. We have added an analysis of the driving mechanisms of land use change in the corresponding section of the original paper, focusing on the regional policy context and socio-economic data. For example, the rapid expansion of construction land use from 1995 to 2000 is closely related to the national urbanization strategy (e.g., the revision of the Land Management Law and the construction policy of development zones) and the average annual growth rate of the regional GDP of 12%, which further supports the synergistic effect between economic and social transformation and the adjustment of land policy. I would like to thank you for your suggestions, which have significantly enhanced the depth of the mechanism explained in this paper! We have added an analysis of the driving mechanism in Section 3.1 (p. 8) of the original text.The specific revisions are as follows:

“continuous economic growth in the TGRR and continuous expansion of urban and rural planning scope, resulting in high demand for building land area and a large amount of development land.

Based on the data analyzed in Table 3, the construction land in the Three Gorges Reservoir Region (TGRR) showed a significant expansion trend from 1995 to 2000, with the area increasing from 434.71 km² to 1,273.61 km², an increase of 193%. This phenomenon is closely related to the national major strategic projects and policy regulation during the same period. Specifically, the Three Gorges Water Conservancy Hub Project was officially launched in 1994, which directly triggered the large-scale resettlement of immigrants in the reservoir area and the acceleration of the urbanization process, prompting a surge in the demand for land for infrastructure, housing and public services. At the same time, the promulgation and implementation of the Regulations on Returning Cultivated Land to Forests in 1998 may guide the conversion of cultivated land and grassland to forest land through the mechanism of ecological compensation, resulting in a 45% reduction in the area of grassland in the same period (2,286.12 km² to 1,408.92 km²), whereas the forest coverage rate remained relatively stable (55.32% to 54.89%), which embodies the dual regulation of the land-use structure by policy. This reflects the dual regulation of the land use structure by the policy. It is worth noting that the water area increased continuously from 964.14 km² in 1995 to 1,573.78 km² in 2020, and the stage-by-stage growth nodes (2003, 2006, and 2008) coincided with the water storage cycle of the Three Gorges Project, which confirms the transformation of the natural geographic pattern by the major projects. 167.78 km² to 34.85 km²) after 2000, which may be related to the binding effect of ecological protection policies such as the National Ecological Functional Zoning (2008).

Fig. 3 Land use status map of the TGRR from 1986 to 2020.Note:The basemap was obtained from the Geospatial Data Cloud(http://www.gscloud.cn/home), and the map boundary has not been changed.Cartographic software:ArcGIS 10.8.”

3.Comment:

The article mentions the trend of carrying capacity during the study period, but does not delve into whether such changes have nonlinear characteristics (such as inflection points or phased changes). The nonlinear trend of carrying capacity changes can be analyzed by regression models.

Response:

We appreciate the reviewers' important suggestions. We fully agree that analyzing the nonlinear characteristics of changes in environmental carrying capacity is crucial to revealing the mechanisms of system evolution. Therefore, we re-examined the data on carrying capacity during the study period to check for nonlinear patterns (e.g., by plotting scatter diagrams and observing the goodness of fit of trend lines). Fitting nonlinear models (such as quadratic curves and exponential models) did not significantly improve the explanatory power of the model.

4.Comment:

Although Table 3, Table 4 and Table 6 contain a large amount of data, they only show numerical changes and lack intuitive visual charts (such as bar charts and line graphs) to enhance understanding. In addition, the map of land use change in Figure 3 lacks legends and text descriptions, and it is suggested to add more clear annotations.

Response:

Thank you for your valuable suggestion. We fully agree with your point of view that intuitive charts are crucial for displaying data trends and changes. In order to present the data trends in Tables 3, 4, and 6 more clearly and enhance their readability and comprehensibility, we have added corresponding trend change charts below or next to each table. The specific modifications are outlined below:

Table 3 Statistical table of land use types and areas in the TGRR from 1986 to 2020 (km2, %).

Land Use Types 1986 1995 2000 2007 2010 2020

Cultivated Land 22141.89 21920.95 21840.97 21699.71 21564.25 21180.44

38.62 38.23 38.09 37.85 37.61 37.22

Forest 31879.08 31720.26 31471.42 31408.35 31380.23 31136.10

55.60 55.32 54.89 54.78 54.73 54.43

Grassland 2306.04 2286.12 1408.92 1405.73 1398.50 1356.62

4.02 3.99 2.46 2.45 2.44 2.43

Water 764.87 964.14 1323.05 1432.63 1484.62 1573.78

1.33 1.68 2.31 2.50 2.59 2.64

Construction Land 234.59 434.71 1273.61 1372.28 1490.63 2071.87

0.41 0.76 2.22 2.39 2.60 3.25

Unused Land 8.80 9.09 17.30 16.57 17.04 16.46

0.02 0.02 0.03 0.03 0.03 0.03

Sum 57335.27 57335.27 57335.27 57335.27 57335.27 57335.27

100 100 100 100 100 100

Table 4 Dynamic degree of land use of various types in the TGRR from 1986 to 2020 (Unit: %).

Period of Time Cultivated Land Forest Grassland Water Construction Land Unused Land Comprehensive Dynamic Degree

1986-1995 -0.11 -0.06 -0.10 2.89 9.47 0.36 0.07

1995-2000 -0.07 -0.16 -7.67 7.45 38.59 18.05 0.42

2000-2007 -0.09 -0.03 -0.03 1.18 1.10 -0.60 0.05

2007-2010 -0.21 -0.03 -0.17 1.21 2.87 0.95 0.09

2010-2020 -0.26 -0.14 -0.04 0.49 6.22 -0.11 0.17

Table 6 Integrated indicator of land use level of various types in the TGRR from 1986 to 2020.

Year Cultivated Land Forest Grassland Water Construction Land Unused Land Regional comprehensiveness △Ib-a

1986 1.1585 1.1120 0.0804 0.0267 0.0164 0.0002 2.3942 0.0031

1995 1.1470 1.1065 0.0797 0.0336 0.0303 0.0002 2.3973 0.0278

2000 1.1428 1.0978 0.0491 0.0462 0.0889 0.0003 2.4251 0.0010

2007 1.1354 1.0956 0.0490 0.0500 0.0957 0.0003 2.4261 0.0017

2010 1.1283 1.0946 0.0488 0.0518 0.1040 0.0003 2.4278 0.0091

2020 1.1166 1.0885 0.0487 0.0528 0.1299 0.0003 2.4369 —

5.Comment:

lthough the article puts forward suggestions on strengthening ecological protection and promoting green development, these suggestions are rather general and lack specific implementation paths. For example, "promoting green development mode" should be combined with the actual situation of the Three Gorges Reservoir area to put forward specific industrial transformation directions or technical application scenarios.

Response:

We sincerely thank you for your valuable comments on the policy recommendations section of this paper! The issue you raised regarding the lack of specific implementation pathways is very important, and it does indeed help to enhance the practical guidance value of the research. Based on your comments, we have focused on deepening and refining the recommendations section in light of the actual characteristics of the Three Gorges Reservoir area. The specific modifications are as follows:

“5.2 Recommendations

In response to the changes in land resources in the TGRR, it is recommended to strengthen land use planning and supervision, with a focus on preserving farmland and ecological land while rationally allocating land for construction. This approach aims to enhance the social, economic, and ecological carrying capacities of the region. Furthermore, promoting green development models, fostering technological innovation, and improving land resource utilization efficiency are crucial steps. Additionally, by refining policies and regulatory frameworks, we can ensure the sustainable utilization of land resources, thereby laying a solid foundation for the region's sustainable development.

In terms of ecological protection, priority should be given to planting flood-tolerant vegetation in the 145–175m elevation zone and adopting stepped ecological slope protection technology to control soil erosion. Concurrently, the basin-wide collaborative governance mechanism should be refined, with the establishment of a Chongqing-Hubei interprovincial water quality monitoring alliance and the creation of a special fund for pollution compensation in the upstream reservoir area. In terms of green development, specific plans for industrial transformation should be outlined: Promote ecological agriculture through “citrus + traditional Chinese medicine” under-forest composite planting; prohibit the construction of new chemical industrial parks and relocate existing enterprises to circular economy transformation zones; implement a subsidy policy for ship LNG power conversion under green shipping initiatives (as outlined in the “Yangtze River Economic Belt Green Development Guidelines”); and implement technical application scenarios: construct distributed photovoltaic power stations in resettlement areas such as Zigui County, and utilize reservoir scheduling AI models to balance power generation with ecological flow. Additionally, new implementation safeguards are introduced, including the establishment of a green industry negative list for reservoir areas and the design of pathways for realizing the value of ecological products.”

6.Comment:

The paper lacks comparative analysis with relevant domestic and foreign studies in the discussion. For example, it can be discussed whether the land use change in the Three Gorges Reservoir area is similar or different from other similar areas (such as the Yellow River basin and the Yangtze River Delta).

Response:

We sincerely appreciate your valuable suggestions! We fully agree that incorporating comparative analyses with similar regions both domestically and internationally in the discussion section will significantly enhance the academic value and practical implications of this research. In response to the reviewers' comments regarding the lack of comparative analysis, we have added systematic domestic and international case comparisons in the discussion section: at the domestic level, we found that the intensity of land use ex

---

## [Decision Letter · Decision Letter 3]

26 Jun 2025

PONE-D-24-37148R3Spatiotemporal Characteristics and Optimization Strategies of Land Use and Land Resource Carrying Capacity in the Three Gorges Reservoir Region(1986–2020)PLOS ONE

Dear Dr. Li,

Thank you for submitting your manuscript to PLOS ONE. After careful consideration, we feel that it has merit but does not fully meet PLOS ONE’s publication criteria as it currently stands. Therefore, we invite you to submit a revised version of the manuscript that addresses the points raised during the review process.

We look forward to receiving your revised manuscript.

Kind regards,

Jun Yang

Academic Editor

PLOS ONE

Journal Requirements:

Additional Editor Comments :

The quality of manuscript have been improved. There still have some problem need to be revised as follow.

1.In keywords, the spatiotemporal characteristics and evaluation could be integrated together, such as evaluation of spatiotemporal characteristics.

2.In introduction, the authors shown three major frontier advances. However, these advances are great, your innovation is not enough compared these advances.

3.The sub-sections are too much. It should be integrated. Such as 3.2,3.3 and 3.4.;3.6 and 3.7. 3.5.1, 3.5.2,3.5.3.

4.3.8 should be moved to discussion.

5.The relevant references should be citied as follow

Spatial and temporal heterogeneity of urban land area and PM2.5 concentration in China. Urban Climate,2022,45:101268. doi: https://doi.org/10.1016/j.uclim.2022.101268.

Challenges and considerations of applying nature-based solutions for furture mega-cities: Implications for Karachi as a Sponge City.Human Settlements and Sustainability,2025,1:50-61. doi: https://doi.org/10.1016/j.hssust.2025.02.002

Reviewers' comments:

Reviewer's Responses to Questions

**Comments to the Author**

1. If the authors have adequately addressed your comments raised in a previous round of review and you feel that this manuscript is now acceptable for publication, you may indicate that here to bypass the “Comments to the Author” section, enter your conflict of interest statement in the “Confidential to Editor” section, and submit your "Accept" recommendation.

Reviewer #3: All comments have been addressed

2. Is the manuscript technically sound, and do the data support the conclusions?

Reviewer #3: Yes

3. Has the statistical analysis been performed appropriately and rigorously? 

Reviewer #3: Yes

4. Have the authors made all data underlying the findings in their manuscript fully available?

Reviewer #3: Yes

5. Is the manuscript presented in an intelligible fashion and written in standard English?

Reviewer #3: Yes

6. Review Comments to the Author

Reviewer #3: The quality of manuscript have been improved. There still have some problem need to be revised as follow.

1.In keywords, the spatiotemporal characteristics and evaluation could be integrated together, such as evaluation of spatiotemporal characteristics.

2.In introduction, the authors shown three major frontier advances. However, these advances are great, your innovation is not enough compared these advances.

3.The sub-sections are too much. It should be integrated. Such as 3.2,3.3 and 3.4.;3.6 and 3.7. 3.5.1, 3.5.2,3.5.3.

4.3.8 should be moved to discussion.

5.The relevant references should be citied as follow

Spatial and temporal heterogeneity of urban land area and PM2.5 concentration in China. Urban Climate,2022,45:101268. doi: https://doi.org/10.1016/j.uclim.2022.101268.

Challenges and considerations of applying nature-based solutions for furture mega-cities: Implications for Karachi as a Sponge City.Human Settlements and Sustainability,2025,1:50-61. doi: https://doi.org/10.1016/j.hssust.2025.02.002

7. PLOS authors have the option to publish the peer review history of their article (what does this mean? ). If published, this will include your full peer review and any attached files.

**Do you want your identity to be public for this peer review?** For information about this choice, including consent withdrawal, please see our Privacy Policy .

Reviewer #3: No

---

## [Author Response · Author response to Decision Letter 4]

27 Jun 2025

Point to Point Responses

Paper Ref. No.:PONE-D-24-37148R3

Title:Spatiotemporal Characteristics and Optimization Strategies of Land Use and Land Resource Carrying Capacity in the Three Gorges Reservoir Region(1986–2020).

Thanks so much for all the kind help to this manuscript. We have analyzed the valuable comments from the editor and reviewers#3 carefully, and tried our best to revise the manuscript. We appreciate the thoughtful review and constructive feedback provided by two viewers.We agree with the reviewer's suggestions and have incorporated the recommended changes into the manuscript.These comments have improved the quality of the paper immensely. The point to point responses are listed as follows.

Comments Editor and Reviewer#3�:

Thanks very much for your comments. According to the following valuable comments and suggestions, we have tried our best to revise the manuscript, and these comments have substantially improved the quality of the paper.

1.Comment:

In keywords, the spatiotemporal characteristics and evaluation could be integrated together, such as evaluation of spatiotemporal characteristics.

Response:

We appreciate the reviewer’s suggestion. Accordingly, we have revised the keywords by integrating “spatiotemporal characteristics” and “evaluation” into a single term: “Evaluation of spatiotemporal characteristics”. This change enhances clarity and better reflects the focus of our study.

“Keywords: Land use; Land Resource Carrying Capacity; Evaluation of spatiotemporal characteristics; Three Gorges Reservoir Region ”

2.Comment:

In introduction, the authors shown three major frontier advances. However, these advances are great, your innovation is not enough compared these advances.

Response:

Thank you for your valuable feedback on our research. We agree with your point that, although we mentioned three cutting-edge developments in the introduction, we need to further emphasize our innovation to highlight our unique contributions compared to these developments. Based on your suggestion, we have added the following content to the introduction to more clearly demonstrate the innovative points and unique value of this study.The specific revisions are as follows:

“In recent years, land use change and its impact on ecosystems and socio-economics have become a global research hotspot. Particularly in the Three Gorges Reservoir Region (TGRR), land use changes have been particularly pronounced due to large-scale infrastructure development and socioeconomic progress. While existing studies have examined land use change trends using multi-source data and long-term time series analysis, assessed land resource carrying capacity through integrated evaluation models, and proposed optimization strategies for sustainable development, these studies often focus on short timeframes or specific types of land changes, lacking systematic integration of multiple influencing factors and concrete, actionable strategies. The innovation of this study lies in: we utilized Landsat imagery and statistical yearbook data from 1986 to 2020 to systematically analyze the spatiotemporal evolution characteristics of land use changes in the Three Gorges Reservoir Region and generated high-precision land use maps; Based on the Analytic Hierarchy Process (AHP) and the mean square deviation decision-making method, we constructed a comprehensive land resource carrying capacity evaluation model that considers social, economic, demographic, and ecological factors; and through an in-depth analysis of the mechanisms by which different driving factors influence land use changes, we proposed targeted optimization strategies aimed at promoting the efficient use of land resources and ecological protection in the Three Gorges Reservoir Area, thereby advancing regional sustainable development.”

3.Comment:

The sub-sections are too much. It should be integrated. Such as 3.2,3.3 and 3.4.;3.6 and 3.7. 3.5.1, 3.5.2,3.5.3.

Response:

We agree with the reviewer’s suggestion regarding the excessive number of subsections. To improve readability and logical flow, we have merged several sections as follows:

Sections 3.2, 3.3, and 3.4 were combined into a new section titled “Land Use Dynamics and Accuracy Assessment in the TGRR”;

Subsections 3.5.1–3.5.3 were integrated into “Verification of land use classification results”;

Sections 3.6 and 3.7 were merged into “Integrated Assessment of Land Resource Carrying Capacity in the TGRR”.

These changes enhance the structural coherence of the methodology section.

4.Comment:

3.8 should be moved to discussion.

Response:

Thank you for this constructive suggestion. We have relocated the content originally presented in Section 3.8 to the Discussion section, where it now serves to interpret the underlying mechanisms and contextualize our findings within broader regional and policy-relevant perspectives.

5.Comment:

The relevant references should be citied as follow

Spatial and temporal heterogeneity of urban land area and PM2.5 concentration in China. Urban Climate,2022,45:101268. doi: https://doi.org/10.1016/j.uclim.2022.101268.

Challenges and considerations of applying nature-based solutions for furture mega-cities: Implications for Karachi as a Sponge City.Human Settlements and Sustainability,2025,1:50-61. doi: https://doi.org/10.1016/j.hssust.2025.02.002

Response:

We thank the reviewer for recommending these relevant references. Both papers have been added to the reference list and appropriately cited in the Discussion section, particularly when discussing spatial heterogeneity and sustainable urban development strategies.

We deeply appreciate the time and effort these reviewers have dedicated to evaluating our manuscript, and we eagerly await any further feedback or suggestions.

---

## [Decision Letter · Decision Letter 4]

1 Aug 2025

Spatiotemporal Characteristics and Optimization Strategies of Land Use and Land Resource Carrying Capacity in the Three Gorges Reservoir Region(1986–2020)

PONE-D-24-37148R4

Dear Dr. Li,

We’re pleased to inform you that your manuscript has been judged scientifically suitable for publication and will be formally accepted for publication once it meets all outstanding technical requirements.

Kind regards,

Jun Yang

Academic Editor

PLOS ONE

Additional Editor Comments (optional):

Accept

Reviewers' comments:

Reviewer's Responses to Questions

**Comments to the Author**

1. If the authors have adequately addressed your comments raised in a previous round of review and you feel that this manuscript is now acceptable for publication, you may indicate that here to bypass the “Comments to the Author” section, enter your conflict of interest statement in the “Confidential to Editor” section, and submit your "Accept" recommendation.

Reviewer #3: All comments have been addressed

Reviewer #4: (No Response)

2. Is the manuscript technically sound, and do the data support the conclusions?

Reviewer #3: Yes

Reviewer #4: Yes

3. Has the statistical analysis been performed appropriately and rigorously? 

Reviewer #3: Yes

Reviewer #4: Yes

4. Have the authors made all data underlying the findings in their manuscript fully available?

Reviewer #3: Yes

Reviewer #4: Yes

5. Is the manuscript presented in an intelligible fashion and written in standard English?

Reviewer #3: Yes

Reviewer #4: Yes

6. Review Comments to the Author

Reviewer #3: All the problems have been addressed. The manuscript have been improved. I think this manuscript could be accepted.

Reviewer #4: The authors made careful revisions in accordance with the reviewers' comments, which addressed my concerns. I have no further comments and recommend this version for publication.

7. PLOS authors have the option to publish the peer review history of their article (what does this mean? ). If published, this will include your full peer review and any attached files.

**Do you want your identity to be public for this peer review?** For information about this choice, including consent withdrawal, please see our Privacy Policy .

Reviewer #3: No

Reviewer #4: No

---

## [Editor Report · Acceptance letter]

PONE-D-24-37148R4

PLOS ONE

Dear Dr. Li,

I'm pleased to inform you that your manuscript has been deemed suitable for publication in PLOS ONE. Congratulations! Your manuscript is now being handed over to our production team.

Kind regards,

on behalf of

Dr. Jun Yang

Academic Editor

PLOS ONE